# Enhanced tumor response to adoptive T cell therapy with PHD2/3-deficient CD8 T cells

Tereza Dvorakova[1,2,3], Veronica Finisguerra[1,2,3], Matteo Formenti[1,2,3], Axelle Loriot[1], Loubna Boudhan [1,2,3], Jingjing Zhu [1,2,3,5] ✉ & Benoit J. Van den Eynde [1,2,3,4,5] ✉

While adoptive cell therapy has shown success in hematological malignancies, its potential against solid tumors is hindered by an immunosuppressive tumor microenvironment (TME). In recent years, members of the hypoxia-inducible factor (HIF) family have gained recognition as important regulators of T-cell metabolism and function. The role of HIF signalling in activated CD8 T cell function in the context of adoptive cell transfer, however, has not been explored in full depth. Here we utilize CRISPR-Cas9 technology to delete prolyl hydroxylase domain-containing enzymes (PHD) 2 and 3, thereby stabilizing HIF-1 signalling, in CD8 T cells that have already undergone differentiation and activation, modelling the T cell phenotype utilized in clinical settings. We observe a significant boost in T-cell activation and effector functions following PHD2/3 deletion, which is dependent on HIF-1α, and is accompanied by an increased glycolytic flux. This improvement in CD8 T cell performance translates into an enhancement in tumor response to adoptive T cell therapy in mice, across various tumor models, even including those reported to be extremely resistant to immunotherapeutic interventions. These findings hold promise for advancing CD8 T-cell based therapies and overcoming the immune suppression barriers within challenging tumor microenvironments.

Adoptive cell therapy involving allogeneic or autologous T cells or genetically engineered T cells is a promising approach that has demonstrated remarkable effectiveness in combating hematological malignancies[1,2]. However, the efficacy of this approach for solid tumors is often limited by lack or loss of tumor-specific antigens and by the immunosuppressive tumor microenvironment (TME), which prevents T cells from infiltrating into the tumor and leads to their exhaustion[3,4]. Several strategies to counteract poor infiltration, survival, and function of adoptively transferred T cells in an immunosuppressive TME have been described[5]. For instance, chimeric antigen receptor (CAR) T cells engineered to express heparinase (HPSE) demonstrated enhanced capacity to degrade the extracellular matrix and improved tumor infiltration and anti-tumor activity[6]. In addition, engineered CAR-T cells expressing high levels of CXCL9 attracted endogenous CD8 T cells to the tumor site and achieved remarkable tumor control in preclinical models[7]. Despite these promising strategies and advancements, significant challenges remain in optimizing adoptive cell therapy for solid tumors.

Even though hypoxia is considered as predominantly immunosuppressive, hypoxia-inducible factor (HIF) has emerged as an important positive regulator of T-cell metabolism and function in recent years[8-12]. Under normoxic conditions, the HIF-α subunit is hydroxylated by prolyl hydroxylase domain-containing enzymes (PHD1-3) and, therefore, recognized by Von Hippel-Lindau (VHL) complex and degraded by the proteasome[13,14]. However, under hypoxia, PHD enzymes are inactive, HIF-α accumulates and, along with

[1]de Duve Institute, UCLouvain, Brussels B-1200, Belgium. [2]Ludwig Institute for Cancer Research, Brussels B-1200, Belgium. [3]WEL Research Institute, Wavre 1300, Belgium. [4]Ludwig Institute for Cancer Research, Nuffield Department of Clinical Medicine, University of Oxford Oxford, Oxfordshire, UK. [5]These authors contributed equally: Jingjing Zhu, Benoit J. Van den Eynde. ✉e-mail: jingjng.zhu@uclouvain.be; benoit.vandeneynde@uclouvain.be

HIF-ß, controls the expression of numerous genes involved in differentiation and effector function of CD8 T cells[15,16]. Particularly, HIF-1α accumulates rapidly after T-cell receptor (TCR) engagement and helps to orchestrate the glycolytic shift crucial for proliferation, survival, and cytokine synthesis of recently activated T cells[17,18].

Several studies involving CD8 T cells isolated from conditional knockout (KO) mice for HIF, PHD, or VHL have demonstrated the potential of stabilizing HIF to enhance T-cell function, offering exciting prospects for therapeutic strategies[9,10,19]. While these findings are encouraging, it is crucial to acknowledge that these studies relied on CD8 T cells sourced from PHD-KO or VHL-KO mouse strains, in which modifications in HIF signaling occur during the whole process of lymphocyte development and differentiation. It remains to be determined whether HIF stabilization also increases effector functions in already differentiated T cells. Moreover, translating such findings into human therapy presents significant challenges. One could think of using pharmacological inhibitors of PHD2/3, but these would have the drawback of upregulating HIF-1α in all cells, including tumor cells, thereby promoting tumor growth[20,21]. It would also stabilize HIF-1α in other immune cells, potentially impairing anti-tumor immunity. Indeed, expression of HIF-1α in macrophages was shown to increase their ability to suppress T-cell function and proliferation[22]. To navigate these complexities, targeting HIF-1α stabilization specifically in CD8 T cells, especially in conjunction with CAR-T or tumor-infiltrating T lymphocytes (TIL) therapy, holds promise, where T cells are typically engineered after differentiation and activation[23]. However, in contrast to previously established findings that suggested HIF-1α stabilization in VHL-KO or PHD-KO mouse models enhances T-cell responses[10,19], the effect of HIF-1α stabilization after CD8 T cell activation on their anti-tumor responses remains uncertain. This uncertainty arises from findings such as those reported by Zhang et al., who observed a delayed growth of B16-OVA tumors in mice who received adoptive transfers of HIF-1α knockdown ovalbumin-specific (OT-1) CD8 T cells[24].

In this study, we stabilize HIF-1α by deleting PHD2/3 in CD8 T cells that are already differentiated and activated, similar to T cells classically used in the clinic for adoptive T-cell therapy. PHD2/3 deletion significantly enhances T-cell activation and effector functions, improving the therapeutic responses to adoptive T-cell transfer in tumor-bearing mice, in various models, including the challenging autochthonous TiRP melanoma model. This effect is dependent on HIF-1α and is accompanied by increased glycolytic flux. These findings hold potential for advancing CD8 T-cell-based therapies and overcoming immune suppression in tumor microenvironments.

## Results

### PHD2/3 deletion in activated CD8 T cells improves tumor response to ACT in induced TiRP melanoma

In order to understand how the modulation of HIF-1α signaling in adoptively transferred CD8 T cells affects their function in vivo when facing an immunosuppressive tumor microenvironment, we modulated the HIF-1α signaling in CD8 T cells and assessed their in vivo anti-tumor function. We took advantage of the TiRP autochthonous inducible melanoma model that expresses the MAGE-type antigen P1A, a model shown to be hypoxic and immunosuppressive[25,26].

Modulation of HIF-1α signaling in tumor antigen-specific CD8 T cells was achieved by nucleofecting activated TCRP1A CD8 T cells with ribonucleoproteins (RNP), consisting of the Cas9 protein in complex with guide RNA (gRNA) targeting HIF-1α or two major isoforms of PHD enzymes expressed by CD8 T cells: PHD2 and PHD3 (Fig. 1a). Guide RNA specificity was validated by qPCR analysis (Suppl Fig. 1A–E) and further confirmed by Western blot and FACS analysis (Fig. 1b and Suppl Fig. 1F). Hypoxia-driven induction of HIF-1α was lost in HIF-1α KO cells, while HIF-1α was clearly stabilized in PHD2/3 KO cells. As depicted in Suppl Fig. 1G, H, the deletion of

PHD2/3 leads to an increase in the expression of two HIF-1α target genes, Slc2a1 and Vegfa. Notably, the impact of PHD2/3 deletion appears to be specific to HIF-1α, with no alterations observed in HIF-2α levels (Suppl Fig. 1I).

We then transferred HIF-1α KO or PHD2/3 KO CD8 T cells intravenously in tumor-bearing TiRP mice, which were shown to be completely resistant to ACT with activated TCRP1A CD8 T cells[25,26]. As anticipated, the administration of ACT using WT (scramble) CD8 T cells did not yield a significant alteration in tumor growth or enhance mice survival, due to the immunosuppressive conditions in the induced TiRP model (Fig. 1c–e). Remarkably, the ACT of PHD2/3 KO CD8 T cells significantly reduced tumor growth. When combined with cyclophosphamide (CTX), a treatment known to reduce immunosuppressive cues and favor the engraftment of newly transferred CD8 T cells[26–28], the ACT of PHD2/3 KO CD8 T cells resulted in an even better anti-tumor response. This combination approach significantly enhanced mice survival (Fig. 1e). In contrast to our expectations, the deletion of HIF-1α in CD8 T cells, on the other hand, did not significantly impact tumor growth (Fig. 1c, d).

These results are intriguing, as hypoxia/HIF-1α have often been associated with suppressed T-cell responses in vivo[24,29–31]. However, our findings indicate an improvement in anti-tumor T-cell function observed when stabilizing HIF-1α through the knockout of PHD2/3 in T cells. We conducted further research to unravel the mechanism by which PHD2/3 KO enhances anti-tumor T-cell function.

### PHD2/3 deletion enhances in vitro CD8 T-cell activation and function

As recently reported[26], exposure of TCRP1A CD8 T cells to a low oxygen environment (1% $O_2$) for a duration of up to 72 h resulted in a significant decrease in both cell survival and proliferation, as illustrated in Fig. 2a, b. This finding aligns with the well-established understanding that hypoxia can impose a suppressive impact on the functionality of immune cells. However, the attempt to rescue the hypoxia-induced decline in cell survival and proliferation through the deletion of HIF-1α yielded unsuccessful outcomes (Fig. 2a, b). Furthermore, we observed no significant changes in CD8 T-cell survival and proliferation when PHD2/3 were deleted. This suggests that alternative hypoxia-induced pathways, independent of HIF, may be contributing to these effects, as demonstrated in Fig. 2a, b.

We conducted additional investigations to assess the influence of HIF-1α on the effector function of CD8 T cells. Initially, we examined the proportion of functional TCRP1A CD8 T cells among CD8 T cells with or without PHD2/3 deletion, defined as those co-expressing Granzyme and Perforin. As illustrated in Fig. 2c, culturing CD8 T cells under hypoxic conditions significantly increases the proportion of effector T cells. However, this increase was completely abolished in CD8 T cells deleted for HIF-1α. Deleting PHD2/3 in CD8 T cells significantly enhanced the percentage of effector CD8 T cells, irrespective of whether they were cultured under normoxia or hypoxia. These results were further supported by qPCR and western blot analysis of Perforin and Granzyme expression (Suppl Fig. 2A, B). Additionally, we confirmed the positive impact of PHD2/3 deletion on the expression of effector molecules also in ovalbumin-specific CD8 T cells (OT-1) (Suppl Fig. 2C).

Furthermore, we detected a consistent increase across various markers associated with T-cell activity in both PHD2/3 knockout and hypoxia-treated T cells, including GITR, CD69, CD137, CTLA-4, PD-1, and LAG3 (Fig. 2d), and confirmed its dependence on HIF-1α. While PD-1 and LAG3 are frequently linked to T-cell exhaustion[32–35], our RT-qPCR analysis on Tox[36,37] and Eomes[38,39], critical regulators of exhaustion features, unveiled a divergent pattern (Fig. 2e).

To ascertain the genuine impact of the HIF-1α pathway on T-cell function, we assessed the cytolytic activity of TCRP1A CD8 T cells

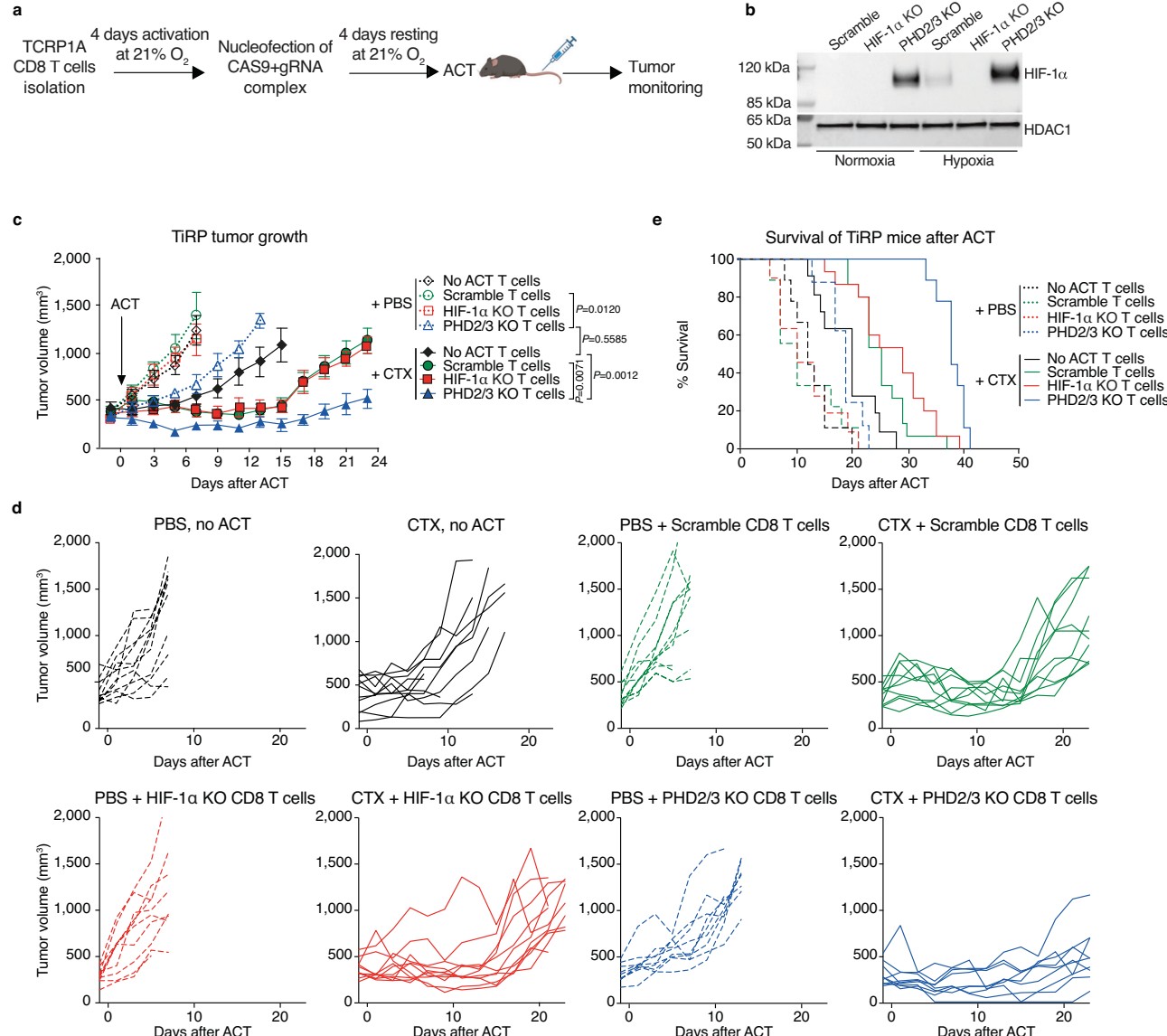

**Fig. 1 | Enhanced tumor rejection upon ACT of CD8 T cells deleted for PHD2/3 in the immunosuppressive TiRP melanoma model. a** Schematic representation of HIF-1α KO and PHD2/3 KO TCRP1A CD8 T cells generation (Created with BioRender.com released under a CC-BY-NC-ND license). **b** Representative western blot analysis showing HIF-1α and HDAC (housekeeping) protein levels in nuclear fractions of scramble, HIF-1α KO, and PHD2/3 KO TCRP1A CD8 T cells incubated under normoxic (21% O₂) or hypoxic (1% O₂) conditions for 24 h. (Representative of three independent experiments). **c** Tumor growth in TiRP mice that have received a single injection of cyclophosphamide (CTX, 100 mg/kg, continuous lines) or PBS (dotted lines) when the tumor size was around 400 mm³, followed or not by ACT of

8 million activated scramble, HIF-1α KO or PHD2/3 KO TCRP1A CD8 T cells 24 h later. (PBS no ACT *n* = 11; PBS + scramble *n* = 11; PBS + HIF-1α KO *n* = 10; PBS + PHD2/3 KO *n* = 9; CTX no ACT *n* = 11; CTX + scramble *n* = 10; CTX + HIF-1α KO *n* = 11; CTX + PHD2/3 KO *n* = 9). **d** Single tumor growth curves for (**C**). **e** Kaplan–Meier survival curve in TiRP mice treated as in (**C**), (PBS no ACT *n* = 12; PBS + scramble *n* = 11; PBS + HIF-1α KO *n* = 11; PBS + PHD2/3 KO *n* = 9; CTX no ACT *n* = 11; CTX + scramble *n* = 15; CTX + HIF-1α KO *n* = 15; CTX + PHD2/3 KO *n* = 9). All data shown are a pool of at least three independent experiments (**a**, **c**–**e**). Data in (**c**) are mean ± SEM. ****p values < 0.0001, calculated by two-way ANOVA with Tukey's multiple comparison correction in (**c**).

against target cells expressing P1A under both normoxic and hypoxic conditions. As illustrated in Fig. 2f, we observed a significant enhancement in cytolytic activity when T cells were cultured in a hypoxic environment, and this enhancement was completely lost when HIF-1α was deleted (Fig. 2f). Furthermore, when T cells were cultured under normoxia but with PHD2/3 knocked out, a similar improvement in cytolytic function was evident. These findings indicated that the stabilization of HIF-1α improved the cytolytic function of CD8 T cells. Notably, PHD2/3 KO in a hypoxic environment even surpassed the lytic activity of hypoxic control cells, further confirming its potential to enhance CD8 T cell activity under both normoxic and hypoxic conditions (Fig. 2f).

To deepen our understanding of the impact of PHD2/3 inhibition in T cells before and after T cell activation, we subjected both naïve and activated CD8 T cells to IOX4, a specific PHD2 inhibitor, and evaluated its effects on the expression of various functional markers. In activated T cells, IOX4 treatment led to elevated expression of Granzyme B, Tox and Lag3 compared to the control group under normoxic conditions (Suppl Fig. 2D). However, unexpectedly, in naïve T cells treated with IOX4, we observed reduced expression of both Granzyme B, Tox and Lag3, suggesting that PHD2 inhibition may have distinct consequences on T-cell function depending on their activation status (Suppl Fig. 2D). These findings supported our approach to inactivate PHD2/3 in activated T cells before ACT.

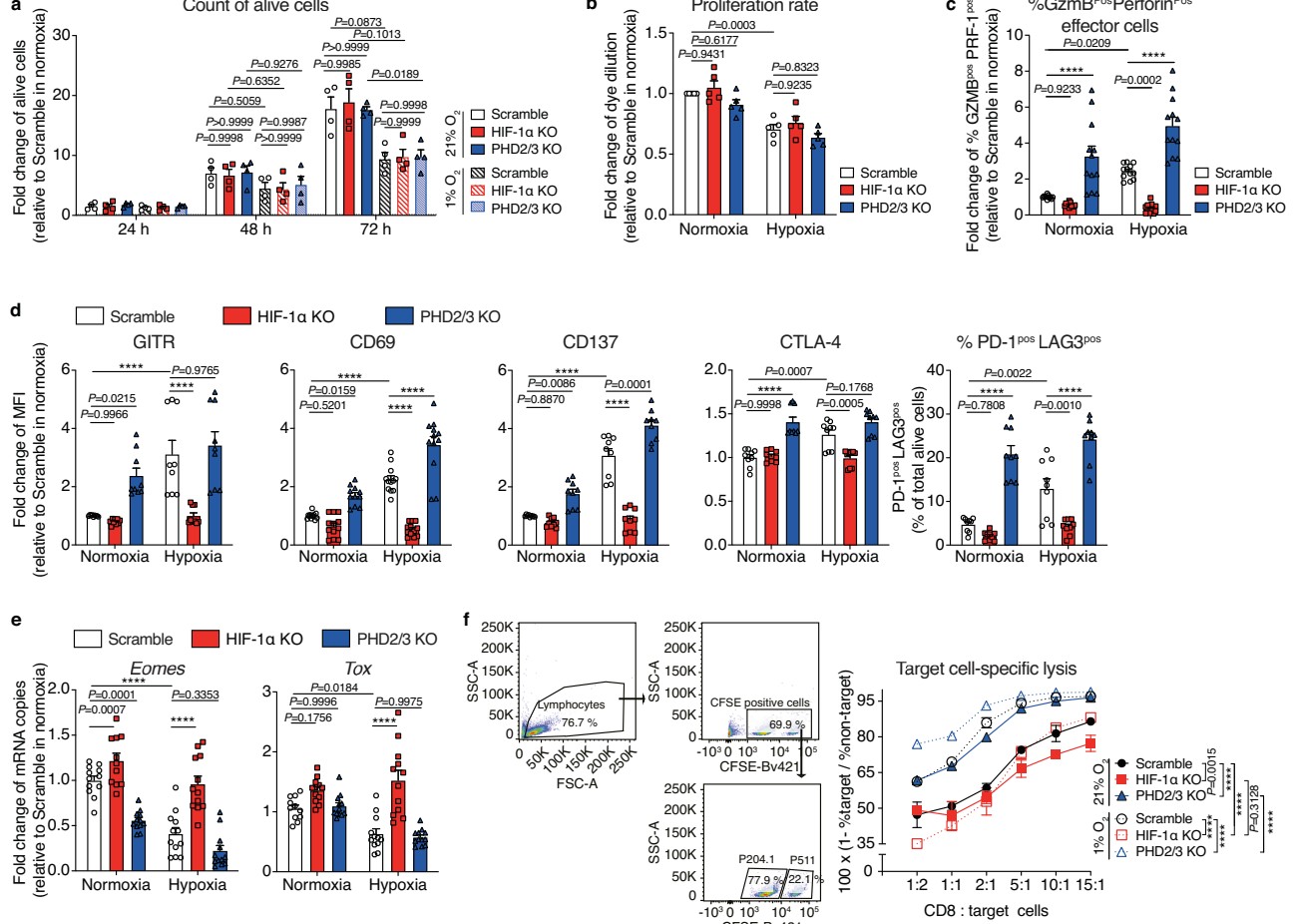

**Fig. 2 | PHD2/3 deletion enhances CD8 T cell effector function. a** Live cell counts of activated scramble, HIF-1α KO and PHD2/3 KO TCRP1A CD8 T cells incubated under normoxic (21% $O_2$) or hypoxic (1% $O_2$) conditions for 72 h. Data were expressed as fold change in cell counts, with normalization to the condition where scramble T cells were cultured under normoxic conditions. **b** Analysis of the proliferation of activated scramble, HIF-1α KO and PHD2/3 KO TCRP1A CD8 T cells incubated in normoxic (21% $O_2$) or hypoxic (1% $O_2$) conditions. Cell proliferation was assessed using CellTrace Violet Cell Proliferation Kit and calculated as dilution of median fluorescent intensity of the violet dye between day 0 (before treatment) and 72 h. Data were expressed as fold change in the dye dilution factor, with normalization to the condition where scramble T cells were cultured under normoxic conditions. **c** Percentage of Granzyme B and Perforin double positive population among total CD8 T cells in the scramble, HIF-1α KO and PHD2/3 KO TCRP1A CD8 T cells incubated under normoxic (21% $O_2$) or hypoxic (1% $O_2$) conditions for 72 h. FACS data were expressed as fold change in the percentage of Granzyme B and Perforin double positive population, with normalization to the condition where scramble T cells were cultured under normoxic conditions. **d** Expression of GITR, CD69, CD137, CTLA-4, PD-1, and LAG3 measured by FACS in cells cultured as in (**c**). Data were expressed as fold change in median fluorescent intensity (MFI), with

normalization to the condition where scramble T cells were cultured under normoxic conditions. **e** RT-qPCR analysis for *Eomes* and *Tox* expression in scramble, HIF-1α KO and PHD2/3 KO TCRP1A CD8 T cells incubated under normoxic (21% $O_2$) or hypoxic (1% $O_2$) conditions for 24 h. The mRNA levels of different genes were measured by quantitative RT-qPCR and normalized to *Actb* (beta-actin). Data were expressed as fold change in mRNA copies, with normalization to the condition where scramble T cells were cultured under normoxic conditions. **f** Killing assay: Activated scramble, HIF-1α KO, or PHD2/3 KO TCRP1A CD8 T cells were cultured under either normoxic (21% $O_2$) or hypoxic (1% $O_2$) conditions for 72 h. Subsequently, these cells were harvested and co-cultured with fluorescently labeled P511 cells (P1A positive, target) and P1.204 cells (P1A negative, non-target) for 5 h, again under either normoxic (21% $O_2$) or hypoxic (1% $O_2$) conditions. The cytotoxicity of CD8 T cells was assessed by determining the proportion of P511 cells (P1A positive) and P1.204 cells (P1A negative) and calculated as 100 × (1 - % P511 cells (target)/% P1.204 cells (non-target)). Data shown as mean ± SEM and are biological replicates (**a**, **b**), or a pool of at least three independent experiments (**c**–**e**), or technical duplicates from one representative out of three independent experiments (**f**). MFI median fluorescence intensity. ****p values <0.0001, calculated by one-way ANOVA with Tukey's multiple comparison test.

## PHD2/3 deletion suppresses in vitro IFN-γ secretion in CD8 T cells

While examining Interferon-γ (IFN-γ) secretion, which serves as an additional indicator of T-cell activation, we came across a rather intriguing observation (Fig. 3a). There was a significant reduction in IFN-γ secretion in T cells undergoing activation when either PHD2/3 deletion was implemented or when the cells were cultured under hypoxic conditions. This finding suggests that the stabilization of HIF-1α exerts an inhibitory effect on IFN-γ secretion, primarily at the protein level, given the entirely contrasting trend observed in *Ifng* mRNA levels (Fig. 3b).

Chang et al. previously uncovered that Glyceraldehyde 3-phosphate dehydrogenase (GAPDH) plays a role in diminishing IFN-γ translation by binding to the 3' untranslated region (3'UTR) of *Ifng*[40]. Expanding upon this, our investigation revealed that activated HIF-1α signaling indeed elevates GAPDH levels in T cells with PHD2/3 deletion or those cultured under hypoxia (Fig. 3c). This increase in GAPDH could potentially account for the inhibitory effect of HIF-1α on IFN-γ production and the contrasting patterns observed between protein and mRNA levels in T cells with activated HIF-1α, though further investigation is warranted to fully elucidate this phenomenon.

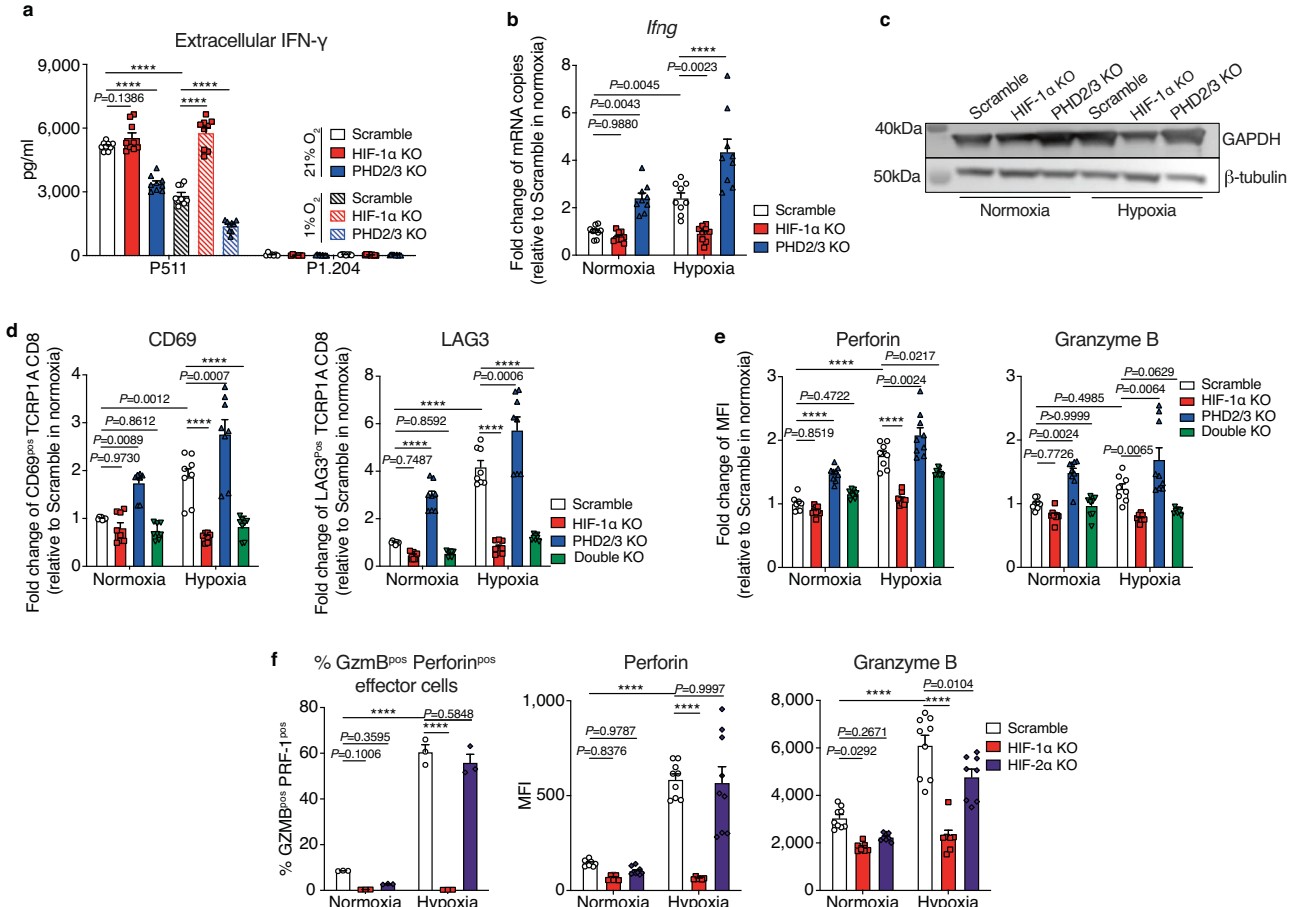

**Fig. 3 | Enhanced effector function of PHD2/3-deleted CD8 T cells is mediated through HIF-1α stabilization. a** Quantification of IFN-γ in the supernatant. scramble, HIF-1α KO, and PHD2/3 KO TCRP1A CD8 T cells were incubated under normoxic (21% O₂), or hypoxic (1% O₂) conditions for 72 h. Subsequently, they were co-cultured with P511 cells (P1A positive) or P1.204 cells (P1A negative) under normoxic (21% O₂) or hypoxic (1% O₂) conditions. After 20 h of co-culture, IFN-γ in the cell supernatant was quantified by ELISA. **b** qPCR analysis of IFN-γ gene expression in activated scramble, HIF-1α KO and PHD2/3 KO TCRP1A CD8 T cells incubated under normoxic (21% O₂) or hypoxic (1% O₂) conditions for 48 h. The mRNA levels of different genes were measured by quantitative RT-qPCR and normalized to *Actb*. Data were expressed as fold change in mRNA copies, with normalization to the condition where scramble T cells were cultured under normoxic conditions. **c** Representative western blot analysis showing GAPDH expression in scramble, HIF-1α KO and PHD2/3 KO TCRP1A CD8 T cells incubated under normoxic (21% O₂) or hypoxic (1% O₂) conditions for 48 h. β-tubulin was used as a housekeeping protein. (Representative of three independent experiments). **d** Flow

cytometry analysis of CD69ᵖᵒˢ and Lag3ᵖᵒˢ population among scramble, HIF-1α KO, PHD2/3 KO, or HIF-1α/PHD2/3 (double) KO TCRP1A CD8 T cells incubated under normoxic (21% O₂) or hypoxic (1% O₂) conditions for 48 h Data were expressed as fold change in cell percentage, with normalization to the condition where scramble T cells were cultured under normoxic conditions. **e** Quantification of Granzyme B and Perforin double positive population by intracellular FACS staining in scramble, HIF-1α KO, PHD2/3 KO, or HIF-1α/PHD2/3 (double) KO TCRP1A CD8 T cells incubated under normoxic (21% O₂) or hypoxic (1% O₂) conditions for 48 h. Data were expressed as fold change in MFI, with normalization to the condition where scramble T cells were cultured under normoxic conditions. **f** Intracellular flow cytometry analysis of the Perforin and Granzyme B double-positive cell population as well as their expression level in the scramble, HIF-1α KO and HIF-2α KO TCRP1A CD8 T cells incubated under normoxic (21% O₂) or hypoxic (1% O₂) conditions for 48 h. Data shown as mean ± SEM and are a pool of at least three independent experiments. MFI median fluorescence intensity. ****p values <0.0001, calculated by one-way ANOVA with Tukey's multiple comparison test.

## PHD2/3 exerts their influence on T-cell function via HIF-1α regulation

Finally, to provide robust support for our findings regarding the improved effector function of PHD2/3 KO CD8 T cells being mediated specifically through the stabilization of HIF-1α, we generated CD8 T cells that had double KO of HIF-1α and PHD2/3 genes. The effects observed in PHD2/3 KO CD8 T cells, including changes in the expression of surface activation markers and effector molecules, were no longer present in these double KO cells (Fig. 3d, e). Furthermore, the deletion of HIF-2α in CD8 T cells did not abolish the induction of Perforin and Granzyme B expression under hypoxic conditions, reinforcing the notion that HIF-1α, rather than HIF-2α, is the key molecule governing the increased T cell function observed in PHD2/3 KO and hypoxic conditions (Fig. 3f)[9].

In summary, our data demonstrate that the deletion of PHD2/3 genes in CD8 T cells substantially boosts their effector function by specifically stabilizing HIF-1α. This enhancement may underlie their increased anti-tumor activity in vivo.

## Deletion of PHD2/3 enhances the in vivo anti-tumor response of CD8 T cells across multiple tumor models

Following our confirmation of the impact of PHD2/3 on T-cell activation, our investigation extended to determining whether stabilizing HIF-1α through deletion of PHD2/3 in CD8 T cells would enhance their anti-tumor function in other tumor models.

Consistent with our observations in the induced TiRP model (Fig. 1c), the transfer of PHD2/3 KO OVA-specific OT-1 T cells resulted in a robust and effective anti-tumor response in the ovalbumin-

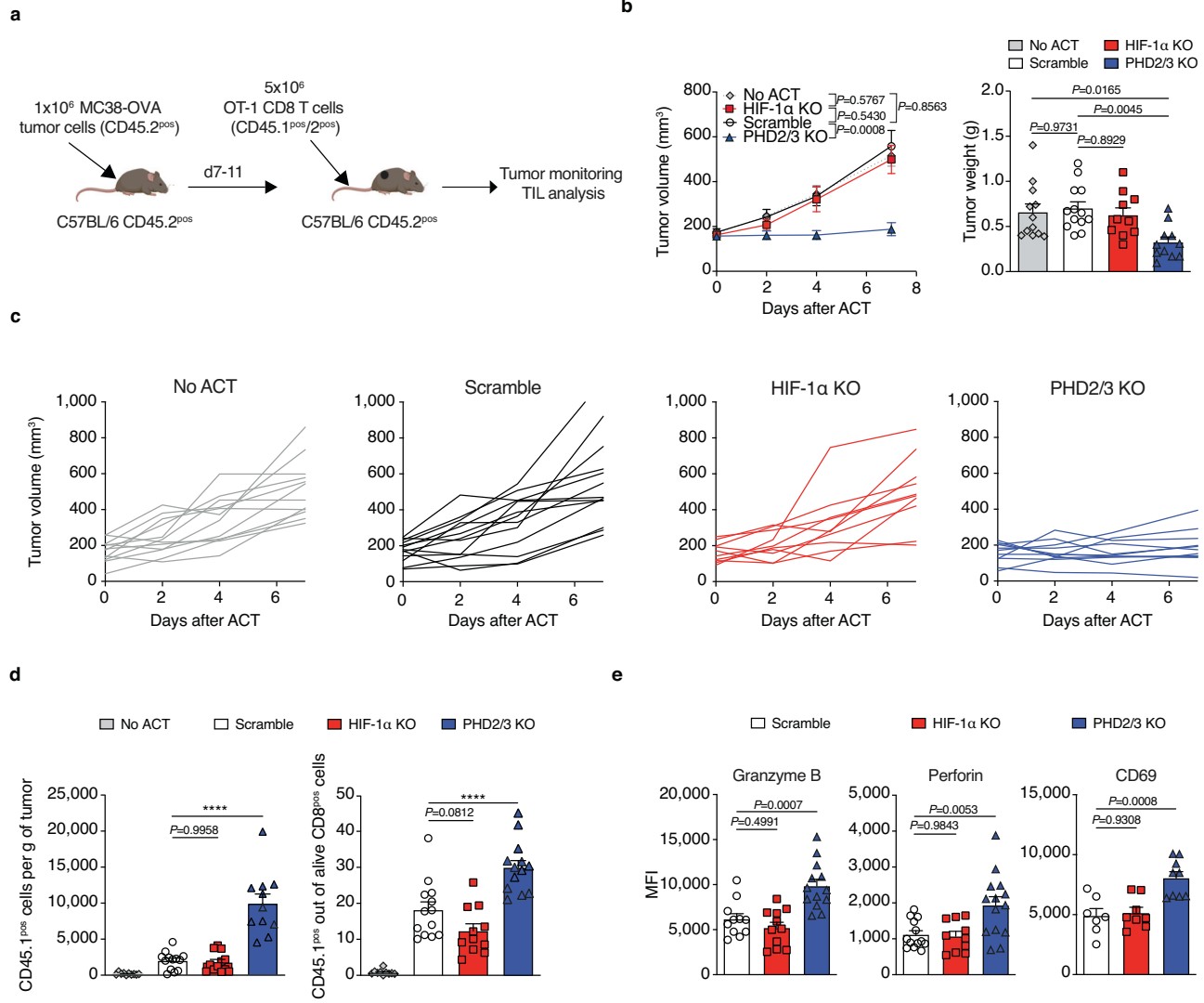

**Fig. 4 | Deletion of PHD2/3 enhances CD8 T cell anti-tumor function via HIF-1α stabilization in the MC38-OVA tumor model. a** Schematic representation of the experimental design. CD45.2 C57/BL6 mice bearing MC38-OVA tumors were treated with CD45.1ᵖᵒˢ CD45.2ᵖᵒˢ OT-1 CD8 T cells (Created with BioRender.com released under a CC-BY-NC-ND license). CD45.1 was used as a marker to differentiate between adoptively transferred (CD45.1ᵖᵒˢ CD45.2ᵖᵒˢ) and endogenous tumor-infiltrating CD8 T cells (CD45.1ⁿᵉᵍ CD45.2ᵖᵒˢ). **b** MC38-OVA tumor growth curve following ACT with 5 million activated scramble, HIF-1α KO or PHD2/3 OT-1 CD8 T cells (left). (tumor growth: no ACT n = 12; scramble n = 13; HIF-1α KO n = 10; PHD2/3 KO n = 11; tumor weight: no ACT n = 12; scramble n = 13; HIF-1α KO n = 10; PHD2/3 KO n = 11). Tumor weight was measured at day 7 (right). **c** Single tumor growth curves for (**b**). **d** Tumor infiltration of transferred OT-1 CD8 T cells, measured by assessing the number of CD8 T cells infiltrating 1 g of tumor (left) or by measuring the proportion of transferred OT-1 CD8 T cells among total CD8 T cells

(right) in MC38-OVA tumors, 3 days after ACT with 5 million activated scramble, HIF-1α KO or PHD2/3 KO CD45.1/2ᵖᵒˢ OT-1 CD8 T cells. (Left: no ACT n = 10; scramble n = 13; HIF-1α KO n = 12; PHD2/3 KO n = 11; right: no ACT n = 10; scramble n = 13; HIF-1α KO n = 11; PHD2/3 KO n = 13). **e** Flow cytometry analysis of Granzyme B, Perforin, and CD69 expression on OT-1 CD8 T cells infiltrating MC38-OVA tumors treated as in (**c**), 3 days after ACT. (Granzyme B: scramble n = 11; HIF-1α KO n = 11; PHD2/3 KO n = 13; Perforin: scramble n = 13; HIF-1α KO n = 10; PHD2/3 KO n = 14; CD69: scramble n = 7; HIF-1α KO n = 8; PHD2/3 KO n = 9). Data were mean ± SEM of biological replicates from one representative experiment out of three independent experiments. MFI median fluorescence intensity. ****p values <0.0001, calculated by one-way ANOVA with Tukey's multiple comparison test (tumor weight **b**), two-way ANOVA with Tukey's multiple comparison test (tumor growth **b**), one-way ANOVA with Tukey's multiple comparison test (**d**, **e**).

expressing MC38 (MC38-OVA) model (Fig. 4a–c), in which activated WT OT-1 CD8 T cells failed to inhibit tumor growth (Fig. 4b). The deletion of HIF-1α in OT-1 CD8 T cells did not exert any significant impact on tumor growth (Fig. 4b), which may be attributed to the observed ineffectiveness of WT OT-1 CD8 T cells in this context. In addition, further investigation of the infiltration of the adoptively transferred CD8 T cells showed that the deletion of PHD2/3 KO in OT-1 CD8 T cells resulted in a notable increase in the infiltration of OT-1 cells within MC38-OVA tumors, as demonstrated in Fig. 4d and Suppl Fig. 3A, three days after the cell transfer. We further analyzed several

markers associated with T-cell function in these tumor-infiltrating OT-1 T cells ex vivo. When compared to scramble T cells, the transferred PHD2/3 KO OT-1 CD8 T cells demonstrated increased expression levels of effector molecules such as Granzyme B, Perforin-1, and the activation marker CD69 (Fig. 4e), indicating enhanced effector function, consistent with our in vitro findings. Although a declined trend was observed, no significant decrease of Granzyme B or Perforin, was found for HIF-1α deleted T cells (Fig. 4e).

Similar results were obtained in the LLC-OVA tumor model (Fig. 5a–d). An improved tumor inhibition, an increased TIL infiltration

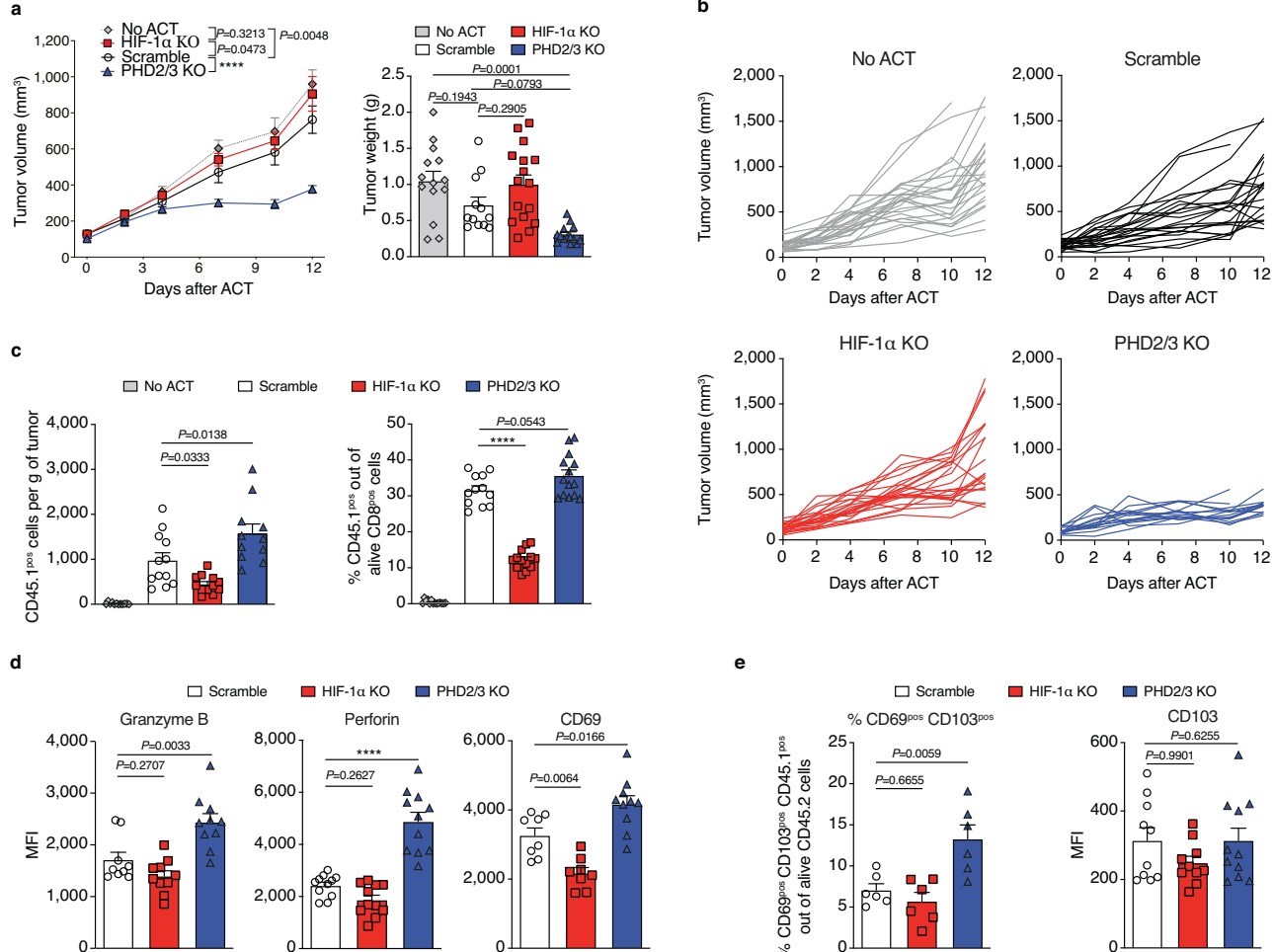

**Fig. 5 | Deletion of PHD2/3 enhances CD8 T cell anti-tumor function via HIF-1α stabilization in the LLC-OVA tumor model. a** LLC-OVA tumor growth curve following ACT of 5 million activated scramble, HIF-1α KO or PHD2/3 KO CD45.1/2[pos] OT-1 CD8 T cells (left) (tumor growth: no ACT $n = 24$; scramble $n = 22$; HIF-1α KO $n = 23$; PHD2/3 KO $n = 15$; tumor weight: no ACT $n = 14$; scramble $n = 12$; HIF-1α KO $n = 16$; PHD2/3 KO $n = 14$). Tumor weight was measured at day 12 (right). **b** Single tumor growth curves for (**a**). **c** Tumor infiltration of transferred OT-1 CD8 T cells, measured by assessing the number of CD8 T cells infiltrating 1 g of tumor (left) or by measuring the proportion of transferred OT-1 CD8 T cells among total CD8 T cells (right) in LLC-OVA tumors, 7 days after ACT with 5 million activated scramble, HIF-1α KO or PHD2/3 KO CD45.1/2[pos] OT-1 CD8 T cells. (Left: no ACT $n = 12$; scramble $n = 12$; HIF-1α KO $n = 12$; PHD2/3 KO $n = 11$; right: no ACT $n = 14$; scramble $n = 12$; HIF-

1α KO $n = 13$; PHD2/3 KO $n = 14$). **d** Flow cytometry analysis of Granzyme B, Perforin, and CD69 expression among transferred OT-1 CD8 T cells in LLC-OVA tumors 7 days after treatment with ACT of 5 million activated scramble, HIF-1α KO or PHD2/3 OT-1 CD8 T cells. **e** Flow cytometry analysis of the proportion of CD69 and CD103 double positive cells as well as their expression level on transferred OT-1 CD8 T cell in LLC-OVA tumors seven days after treatment with ACT of 5 million activated scramble, HIF-1α KO or PHD2/3 OT-1 CD8 T cells. Data were mean ± SEM of biological replicates from one representative experiment out of three independent experiments. MFI median fluorescence intensity. ****$p$ values <0.0001, calculated by two-way ANOVA with Tukey's multiple comparison test (**a** left panel), or one-way ANOVA with Tukey's multiple comparison test (**a** right panel, **c**–**e**).

and expression of granzyme B, perforin, as well as CD69 were found in mice treated with PHD2/3 KO CD8 T cells, when analyzed 7 days after ACT. These findings collectively underline the potential of stabilizing HIF-1α through PHD2/3 KO to enhance T-cell tumor infiltration and improve the overall efficacy of immunotherapy.

In addition, in line with findings reported by Liikanen et al, a higher proportion of CD69[pos] CD103[pos] OT-1 CD8 T cells was found in tumors of mice treated with PHD2/3 KO OT-1 CD8 T cells (Fig. 5e), indicating an increase of tumor resident memory (Trm) subset of CD8 T cells[19]. However, in our case, the expansion of the CD69[pos] CD103[pos] subset in PHD2/3 KO CD8 T cells was driven solely by an increase in CD69 expression, and no alteration was found for CD103 (Fig. 5e)[19]. Further investigation into stem-like markers Slamf6 and TCF1 expression on transferred T cells in MC38-OVA tumors revealed a notable difference between PHD2/3 KO and scramble CD8[pos] T cells. Specifically, there was a higher percentage of Slamf6-positive CD8[pos] T cells among the transferred T cells in mice that received PHD2/3 KO T cells

compared to those that received scramble T cells (Suppl Fig. 3B). A similar pattern was observed for TCF1, another marker widely used to define "stem-like" or precursor T cells. In line with these phenotypic changes, tumor rechallenge experiments demonstrated a better long-term protective effect of PHD2/3 KO OT-1 T cells as compared to control OT-1 T cells (Suppl Fig. 3C) in the MC38-OVA tumor model. In these experiments, in which simultaneous tumor implantation and adoptive cell transfer were conducted, ~40% of mice treated with scramble T cells developed tumors, vs none in the PHD2/3 KO group. We then subjected tumor-free mice from both groups to an identical tumor cell rechallenge 4 weeks later. The scramble group demonstrated a tumor recurrence rate of ~33%, whereas the PHD2/3 KO group exhibited a mere 5% recurrence rate (Suppl Fig. 3C). Furthermore, upon rechallenging mice with intravenous injections of MC38-OVA tumor cells, a notable increase in the expansion of transferred CD8 T cells was observed in the blood in the group that received adoptive cell therapy (ACT) of PHD2/3 KO OT-1 T cells compared to the control

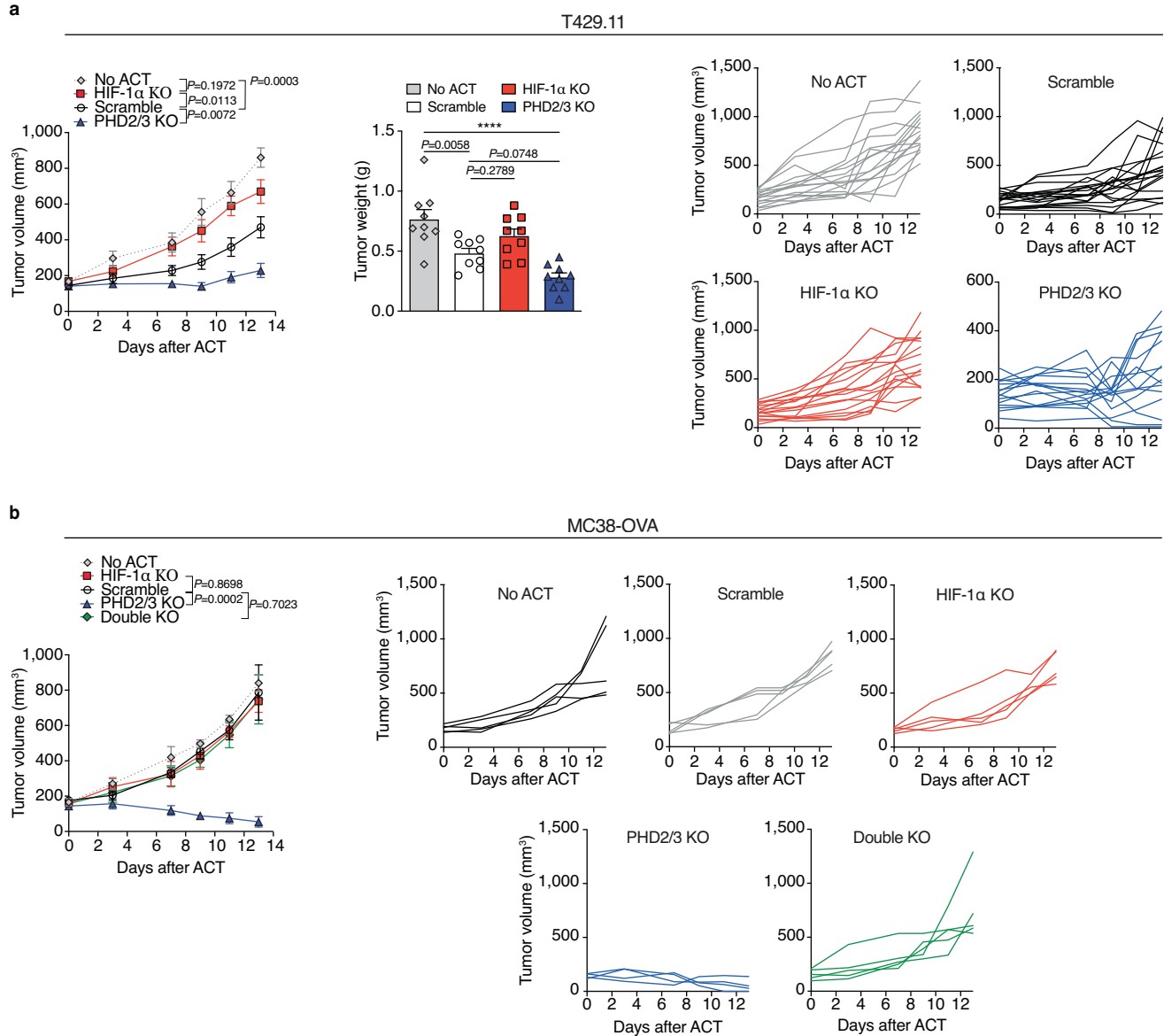

**Fig. 6 | The impact of HIF-1α or PHD2/3 deletion in CD8 T cells on their anti-tumor efficacy. a** The growth curve of T429.11 tumors following treatment with ACT of 8 million activated Scramble, HIF-1α KO or PHD2/3 TCRP1A CD8 T cells (left). (tumor growth: no ACT $n = 16$; scramble $n = 19$; HIF-1α KO $n = 16$; PHD2/3 KO $n = 15$; tumor weight: no ACT $n = 9$; scramble $n = 9$; HIF-1α KO $n = 9$; PHD2/3 KO $n = 9$). Tumor weight was measured at day 13 (middle). Single tumor growth curves (right). **b** The growth curve of MC38-OVA tumors following treatment with scramble, HIF-

1α KO, PHD2/3 KO, or HIF-1α/PHD2/3 double KO CD45.1/2$^{pos}$ OT-1 CD8 T cells (left, $n = 5$). Single tumor growth curves (right). Data were mean ± SEM of biological replicates from one representative experiment out of three independent experiments. ****$p$ values <0.0001, calculated by one-way ANOVA with Tukey's multiple comparison test (tumor weight **a**), two-way ANOVA with Tukey's multiple comparison test (tumor growth **a**, **b**).

group (Suppl Fig. 3D). This finding provides additional confirmation of the superior "memory" or "progenitor" phenotype exhibited by the PHD2/3 KO T cells.

**The enhanced in vivo anti-tumor response of CD8 T cells following deletion of PHD2/3 is mediated through HIF-1α signaling**
The influence of HIF-1α stabilization on the effectiveness of anti-tumor CD8 T cells was further affirmed using the TiRP-derived T429.11 melanoma mouse model, which expresses the P1A antigen (Fig. 6a) and exhibits a partial response to the ACT of TCRP1A CD8 T cells[25]. Again, TCRP1A CD8 T cells lacking PHD2/3 demonstrated the most pronounced anti-tumor capacity, as shown in Fig. 6a. In contrast, deletion of HIF-1α in TCRP1A CD8 T cells showed attenuated anti-tumor activity,

confirming the pivotal role of HIF-1α signaling in the modulation of CD8 T-cell responses against tumors.

To have final confirmation of the role of HIF-1α in mediating the enhanced anti-tumor CD8 T-cell function in PHD2/3 KO CD8 T cells, we generated HIF-1α and PHD2/3 double knockout (double KO) OT-1 CD8 T cells and administered them to mice with MC38-OVA tumors. As shown in Fig. 6b, the increased tumor inhibition by PHD2/3 KO CD8 T cells was completely abrogated in mice treated with double KO CD8 T cells.

In conclusion, our study provides strong evidence supporting the significance of HIF-1α stabilization through PHD2/3 deletion within CD8 T cells as an effective approach to enhance the anti-tumor response of CD8 T cells.

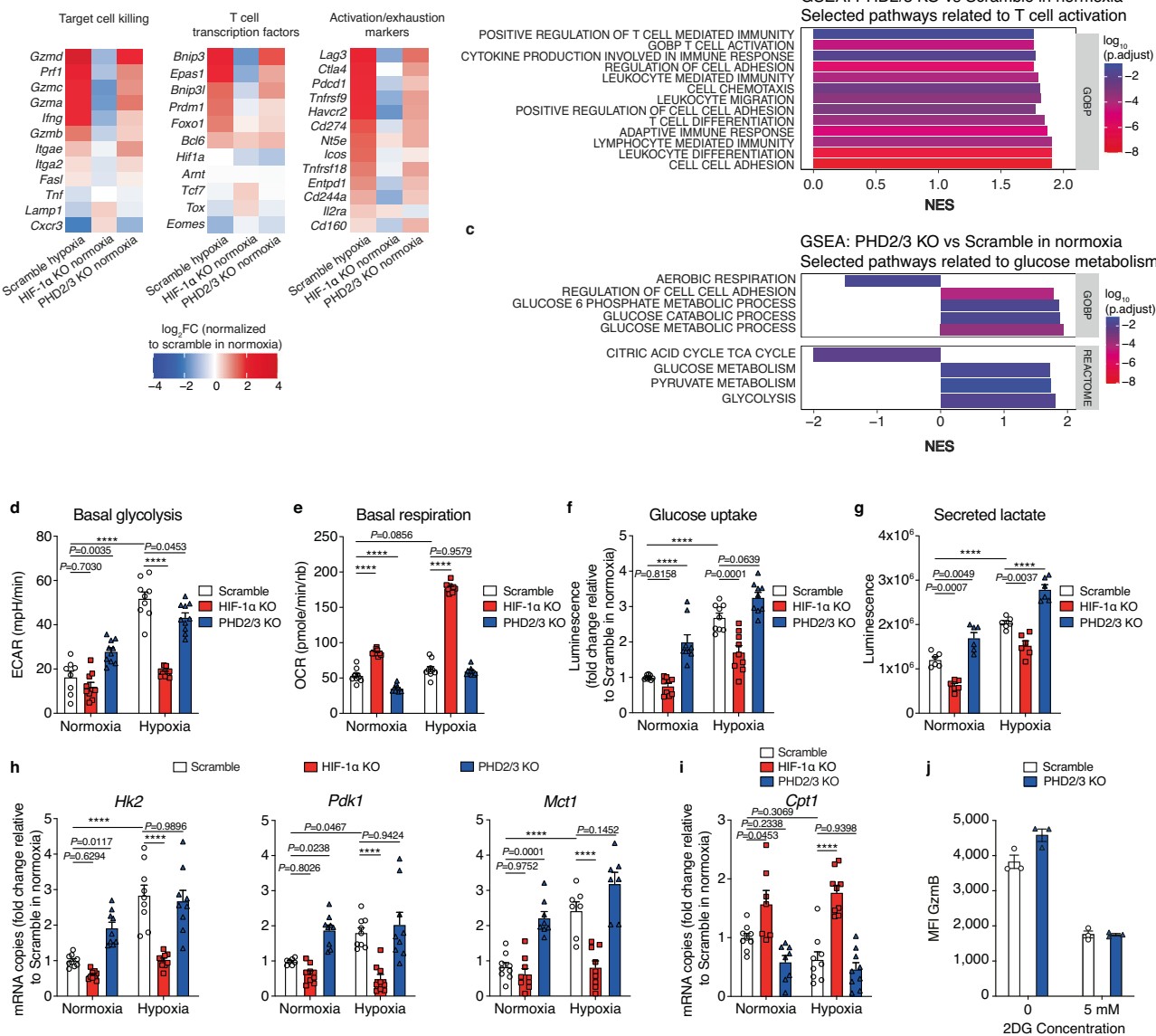

**Fig. 7 | Deletion of PHD2/3 increases glycolytic flux in CD8 T cells. a** RNA sequencing analysis was conducted on activated scramble, HIF-1α KO, and PHD2/3 KO TCRP1A CD8 T cells following incubation under normoxic (21% O₂) or hypoxic (1% O₂) conditions, each for a duration of 24 h. The tables depict the log₂ fold change (log₂FC) in the expression level of selected genes, relative to scramble TCRP1A CD8 T cells cultured under normoxic conditions. **b, c** Gene enrichment analysis of PHD2/3 KO TCRP1A CD8 T cells compared to scramble TCRP1A CD8 T cells cultured under normoxic conditions. NES: Normalized enrichment score. **b** Selected pathways related to T-cell activation. **c** Selected pathways related to glucose metabolism. **d, e** Activated scramble, HIF-1α KO, and PHD2/3 KO TCRP1A CD8 T cells were incubated under normoxic (21% O₂) or hypoxic (1% O₂) conditions for a duration of 72 h. Extracellular acidification rate (ECAR) (**d**) and oxygen consumption rate (OCR) (**e**) were measured using a Seahorse XF Analyzer. **f** Measurements of glucose uptake in cells treated as in (**d, e**). **g** Measurement of extracellular lactate in supernatants of cells treated as in (**d, e**). **h** RT-qPCR analysis

for hexokinase 2 (*Hk2*), pyruvate dehydrogenase kinase (*Pdk1*), and mono-carboxylate transporter 1 (*Mct1*) in scramble, HIF-1α KO and PHD2/3 KO TCRP1A CD8 T cells incubated under normoxic or hypoxic conditions for 24 h. **i** RT-qPCR analysis for Carnitine palmitoyltransferase (*Cpt1*) in scramble, HIF-1α KO and PHD2/3 KO TCRP1A CD8 T cells incubated under normoxic or hypoxic conditions for 24 h. For (**h, i**), the mRNA levels of different genes were measured by quantitative RT-qPCR and normalized to *Actb*. Data were expressed as fold change in mRNA copies, with normalization to the scramble T cells cultured under normoxic conditions. **j** Flow cytometry analysis of Granzyme B expression on a scramble or PHD2/3 KO OT-1 CD8 T cells treated or not with 5 mM 2-deoxy-ᴅ-glucose (2-DG). Data in (**d**–**i**) are shown as mean ± SEM and are a pool of at least three independent experiments. **j** is shown as mean ± SEM and is a representative experiment out of three independent experiments. ****p values <0.0001, calculated by one-way ANOVA with Tukey's multiple comparison test.

## Enhanced effector function of PHD2/3 KO CD8 T cells is associated with increased glucose metabolism

To decipher mechanisms behind the enhanced anti-tumor efficacy of TCRP1A CD8 T cells knocked out for PHD2/3 or cultured under hypoxia, we analyzed and compared the transcriptome of activated TCRP1A CD8 T cells with HIF-1α KO or PHD2/3 KO exposed to different oxygen conditions (21% O₂ or 1% O₂) for 48 h (Fig. 7a–c).

Our RNAseq analysis supports our functional findings. When WT CD8 T cells were exposed to hypoxia, key genes for T-cell function showed increased expression (Fig. 7a). Deleting PHD2/3 in CD8 T cells had a similar effect, mimicking the gene expression changes induced by hypoxia. Notably, deleting HIF-1α in CD8 T cells didn't cause significant changes. Gene set enrichment analysis (GSEA) confirmed that deleting PHD2/3 enhances pathways associated with T-cell function, as

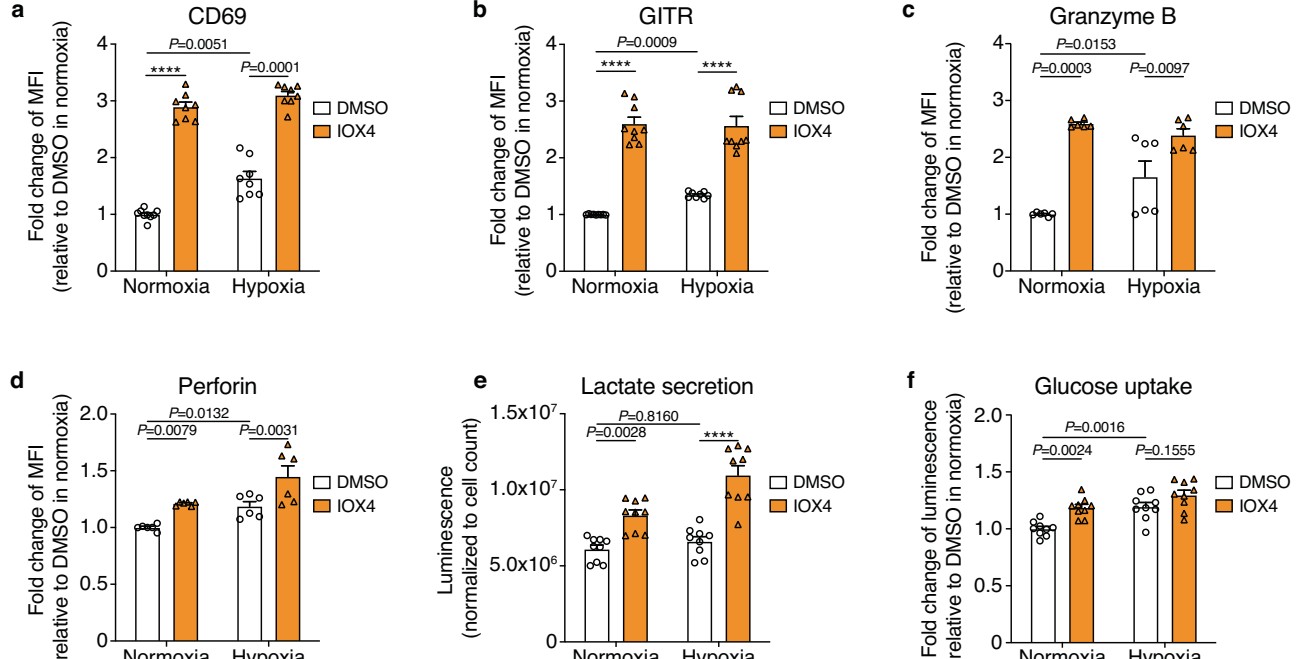

**Fig. 8 | Pharmacological inhibition of PHD enzymes with IOX4 improves human CD8 T-cell function. a–d** Flow cytometry analysis for CD69 (**a**), GITR (**b**), Granzyme B (**c**), and Perforin (**d**) expression on activated human CD8 T cells treated or not with IOX4 (20 μM), and incubated under normoxic (21% $O_2$) or hypoxic (1% $O_2$) conditions for 48 h. Data were expressed as fold change in MFI, with normalization to untreated CD8 T cells cultured under normoxic conditions. **e** Quantification of lactate in the supernatant of activated human CD8 T cells treated or not with IOX4 and incubated under normoxic (21% $O_2$) or hypoxic (1% $O_2$) conditions for 24 h. **f** Glucose uptake measurement on activated human CD8 T cells treated as in (**e**). Data were expressed as fold change in luminescence, with normalization to untreated CD8 T cells cultured under normoxic conditions. Data were shown as mean ± SEM and are a pool of at least three independent experiments. MFI median fluorescence intensity. ****$p$ values <0.0001, calculated by one-way ANOVA with Tukey's multiple comparison test.

demonstrated in Fig. 7b. In addition, the deletion of PHD2/3 in CD8 T cells enriched pathways related to glucose metabolism (Fig. 7c).

It has been extensively demonstrated that activated CD8 T cells rely primarily on glycolysis to sustain their proliferation and effector function[40–42]. To see whether the enhanced function of PHD2/3 KO CD8 T cells was associated with HIF-1α-mediated increase in glucose metabolism, we performed a metabolic analysis on WT, HIF-1α KO, and PHD2/3 KO CD8 T cells cultured under normoxic or hypoxic conditions. Our findings indicate that WT CD8 T cells cultured under hypoxic conditions exhibit higher extracellular acidification rate (ECAR), indicating an elevated glycolytic flux in these cells (Fig. 7d). The same was found for CD8 T cells deleted for PHD2/3 cultured under normoxic or hypoxic conditions. A decreased OCR was observed in CD8 T cells with PHD2/3 deletion under normoxic conditions (Fig. 7e). HIF-1α knockout reduced ECAR under hypoxic conditions and increased OCR in both normoxia and hypoxia (Fig. 7d, e). Together, these results indicate that HIF-1α stabilization, mediated either by hypoxia or by PHD2/3 knockout, enhances glycolysis.

In line with these observations, we observed consistent increases in glucose uptake (Fig. 7f), extracellular lactate (Fig. 7g), and expression of key glycolytic enzymes such as Hexokinase 2 (*Hk2*) and phosphoinositide-dependent kinase-1 (*Pdk1*), along with the mono-carboxylate transporter 1 (*Mct1*), in CD8 T cells cultured under hypoxic conditions and PHD2/3 KO CD8 T cells cultured in both normoxic and hypoxic conditions (Fig. 7h). Conversely, the expression of carnitine palmitoyltransferase (*Cpt1*), the rate-limiting enzyme in fatty acid oxidation, appeared to be negatively regulated by HIF-1α in CD8 T cells (Fig. 7i). The involvement of glycolysis in the increased effector functions of PHD2/3 KO CD8 T cells was supported by the observation that inhibiting glycolysis using 2-deoxy-D-glucose (2-DG) abolished the increased Granzyme B expression induced by PHD2/3 KO (Fig. 7j).

Altogether, these data provide further support for the notion that the enhanced effector functions observed in PHD-KO CD8 T cells are linked to an increased glycolytic flux[10].

## HIF-1α stabilization boosts effector functions of human CD8 T cells

Finally, to assess the translational implications of our findings, we explored the effects of HIF-1α stabilization through pharmacological inhibition of PHD enzymes on human CD8 T cells.

We treated activated polyclonal human CD8 T cells with a potent PHD2 inhibitor known as IOX4. After 24 h of IOX4 treatment, we observed significant increases in activation markers like GITR and CD69, as well as effector molecules such as Granzyme B and Perforin, both in normoxic and hypoxic conditions when compared to the control group (Fig. 8a–d). Additionally, IOX4 treatment also resulted in elevated lactate secretion and increased glucose uptake, indicative of a higher glycolytic rate (Fig. 8e, f). Even though we do not envision systemic treatment with PHD inhibitors due to opposite effects on other cells, these data validate the notion that HIF-1α stabilization also increased effector functions in human CD8 T cells. Collectively, these findings mirror our observations with mouse CD8 T cells and suggest that the stabilization of HIF-1α in human T cells before ACT could offer a promising strategy to sustain T-cell effector functions, potentially leading to improved clinical outcomes of T-cell therapies.

## Discussion

While adoptive cell therapies show promise, there is an urgent need to enhance their efficacy. Here, we propose a strategy for improving the survival and effector functions of adoptively transferred CD8 T cells by genetically or pharmacologically activating the HIF-1α pathway in CD8 T cells before their infusion into patients. We have used CD8 T cells

targeting OVA as a model antigen, but also P1A, a MAGE-type antigen, as a more relevant natural tumor antigen. We demonstrate that the genetic knockout of PHD2/3 in activated CD8 T cells prior to adoptive cell transfer significantly improves their survival and efficacy in inhibiting tumor growth in vivo, across a range of tumor models. This innovative approach holds the potential to significantly elevate the therapeutic outcomes of cell-based therapies.

Recently, several research groups have directed their efforts toward stabilizing HIF-1α, specifically in lymphocytes, to enhance tumor control. Stabilization of HIF-1α in both CD4 and CD8 lymphocytes (utilizing CD4 Cre PHD2[fl/fl] mice) led to improved control of EG7-OVA tumors[10]. In line, adoptively transferred VHL-deficient OT-1 CD8 T cells differentiated into highly cytotoxic tissue-resident memory T-cells and exhibited superior control over B16-OVA tumors compared to their WT counterparts[19]. Although promising, it is important to recognize that these studies utilized CD8 T cells isolated from conditional PHD-KO or VHL-KO mice, resulting in alterations in HIF signaling during the development of lymphocytes.

Considering the intricate nature of HIF-1α-dependent metabolic reprogramming throughout various T-cell states, it is plausible that the manipulation of HIF-1α signaling in different stages may yield differing effects on T-cell fitness. Indeed, a study involving the adoptive transfer of HIF-1α knockdown CD8 T cells has yielded conflicting results regarding the role of HIF signaling in the anti-tumor activity[24]. In line, CD8[+] T cells exposed to hypoxia shortly after activation exhibited impaired elevation of CD25 expression and decreased proliferative capacity[43]. These impairments were partially mitigated when cells were exposed to hypoxia at a later time point following activation[43]. To gain insight into this, we explored the consequences of targeting HIF-1α in CD8 T cells in the context of therapeutic adoptive cell transfer, in which well-differentiated and mature T cells are used.

We show that despite HIF-mediated induction of immune checkpoint molecules such as CTLA-1, PD-1 and LAG3, the deletion of PHD2/3 in activated CD8 T cells resulted in an upregulation of activation markers and effector molecules, and ultimately led to an enhanced cytolytic capacity of CD8 T cells. Furthermore, adoptively transferred PHD2/3 KO CD8 T cells better controlled tumor growth in several models, including the highly immunosuppressive model of autochthonous melanoma TiRP.

Our study also revealed that the increased effector function of PHD2/3 KO CD8 T cells was linked to an increase in glucose metabolism. This finding aligns with the established understanding that HIF signaling plays a pivotal role in orchestrating the metabolic transition linked to the acquisition of T cell effector functions[8,44].

Intriguingly, in line with several publications, we show that hypoxia/HIF-1α seems to negatively regulate the secretion of IFN-γ by CD8 T cell upon restimulation, despite increasing transcription of *Ifng* mRNA[26,45,46]. We propose that this is due to increased GAPDH in hypoxic conditions or when HIF-1α is stabilized by PHD2/3 deletion. Indeed, GAPDH, a well-known HIF-1α target gene[47], was shown to reduce the translation of IFN-γ by binding to AU-rich elements within the 3′ untranslated region (UTR) of *Ifng* mRNA in mouse CD4 T cells[40]. Interestingly, cytotoxic genes such as granzymes and perforin do not possess these AU-rich elements and are, therefore, not directly regulated by glycolytic enzymes[48]. These insights offer a compelling perspective on the complex interplay between HIF-1α, glycolytic enzymes, and cytokine production in CD8 T cells.

Furthermore, while both HIF-1α and HIF-2α are stabilized in CD8 T cells upon activation, the involvement of HIF-2α in T cell function has not been well established[9]. On one hand, it was described that CD8 T cells lacking HIF-2α didn't display any changes in glycolytic metabolism or activation[9]. On the other hand, an over-expression of HIF-2α in CD8 T cells, as opposed to HIF-1α, was found to boost the anti-tumor cytotoxicity of CD8 T cells[49]. By using a controlled way of deleting HIF-1α, HIF-2α, and PHD2/3 in CD8 T cells, we could rule out the involvement of HIF-2α in CD8 T cells in the context of enhanced T cell function induced by PHD2/3 KO.

Finally, it's worth noting that we observed an increase in the cytolytic potential of human CD8 T cells in vitro when treated with IOX4, a pharmacological inhibitor of PHD enzymes, mirroring our observation with mouse CD8 T cells deleted for PHD2/3 KO. Whether this pre-treatment is sufficient to confer long-term tumor control after ACT still needs to be investigated. An encouraging study demonstrated that conditioning and culturing CD8 T cells under hypoxic conditions prior to their adoptive transfer substantially boosted their cytolytic potential, resulting in a more potent anti-tumor response in a murine cancer model[50].

While our findings suggest promising prospects for leveraging PHD2/3 to enhance T-cell therapy, it is crucial to consider the broader implications of PHD2/3 inhibition. A systemic inhibition of PHD2/3 with pharmacological agents may exert unintended effects on other cell types due to the ubiquitous expression of the PHD/HIF system. Of particular concern is the role of HIF-1α in tumor progression and metastasis, which requires cautious consideration in therapeutic interventions targeting the PHD2/3- HIF-1α axis[20,21,51]. Another concern is the effect on other immune cells, such as macrophages, in which an HIF-1α-mediated hypoxia response was shown to increase T-cell suppression[22]. In response to these concerns, our proposed strategy focuses on selectively inhibiting PHD2/3 activity within CD8 T cells in the context of ACT, thereby avoiding potential off-target effects while enhancing T-cell function. This targeted approach holds promise for optimizing the efficacy and safety of T cell-based immunotherapies and warrants further investigation in preclinical and clinical settings, not only for TCR-based therapy as discussed here, but also potentially for CAR-T cell therapies.

In summary, within our study, two key aspects of HIF signaling have come to light. First, our research underlined the intricate and multifaceted impact of HIF-1α signaling on CD8 T cells. Second, HIF-1α stabilization by PHD2/3 inactivation in activated CD8 T cells prior to their adoptive transfer increases their anti-tumor activity and appears as a promising approach to improve cancer immunotherapies.

## Methods
### Mice
This research complies with all relevant ethical regulations. Animal welfare rules were followed according to the 2010/63/EU Directive, with all procedures approved by the Animal Ethical Committee of UCLouvain University (references 2015/UCL/MD/15 and 2019/UCL/MD/018). B10.D2 TiRP-10B[+/+];Ink4a/Arf[flox/flox] (TiRP) mice, B10.D2;Ink4a/Arf[flox/flox] (TiRP-neg) and B10.D2-Rag1KO mice heterozygous for the H-2L[d]/P1A[35-43]-specific TCR transgene (TCRP1A) were previously described[25]. C57BL/6 mice were purchased from Harlan. CD57BL/6-Tg(Tcra Tcrb)1100Mjb/Crl (OT-1-OVA-TCR) were bred with B6.SJL-Ptprc[a] Pepc[b]/BoyCrl mice (both purchased from Charles River) to generate heterozygous CD45.1[pos] and CD45.2[pos] OT-1-OVA-TCR mice. Female mice (aged 9–12 weeks) were used for CD8 T cell isolation and for in vivo tumor transplantation to avoid the rejection of male-specific antigens that could occur when using male mice. The sample size was chosen based on preliminary experiments and previously published results. All mice were produced under specific-pathogen-free conditions at the de Duve Institute, fed with RM1 diet (SDS801002), housed at an ambient temperature of around 21–23 °C with 40–60% humidity and a 12 h light–dark cycle. The mice were euthanized before their tumors reached a humane endpoint (Tumor Volume less than 2000 mm³).

### Cell lines
L1210.P1A.B7-1, T429.11, P511, and P1.204 cell lines were described previously[25]. LL2-Thy1.1-OVA (LLC-OVA) cells, expressing a cytoplasmic form of ovalbumin, were a kind gift from D. Fearon (Cambridge).

MC38-OVA cells expressing a cytoplasmic form of ovalbumin were generated as described previously[52].

All cells were maintained at 37 °C and 8% CO$_2$. All culture media contained 10% FBS, L-arginine (0.55 mM, Merck), L-asparagine (0.24 mM, Merck), Glutamine (1.5 mM, Merck), 50 U/mL penicillin, and 50 mg/mL streptomycin (Life Technologies) (complete medium). Cell lines were routinely tested for mycoplasma contamination. For experiments in hypoxia, a hypoxystation (Whitley H35) at 37 °C, 8% CO$_2$ and 1% O$_2$ was used.

## Mice treatments
4 OH-Tamoxifen (Imaginechem) dissolved in Ethanol and mineral oil (1:9 ratio) and sonicated for 30 min, was injected subcutaneously (2 mg/200 μL) in the neck area of female TiRP mice 7–9 weeks old. About 7–9 weeks old CD57BL/6 WT female mice were injected subcutaneously with 10$^6$ LLC-OVA or MC38-OVA cells. Female 7–9 weeks old TiRP-neg mice were injected subcutaneously with 10$^6$ T429.11 cells. Tumor volume was calculated as Volume = π × width$^2$ × length/6. Before treatment, mice were randomized based on tumor size. Cyclophosphamide (Sigma, C7397) was injected intraperitoneally at 100 mg/kg. For ACT, 8 × 10$^6$ activated and nucleofected TCRP1A CD8 T cells or 5 × 10$^6$ activated and nucleofected OT-1 CD45.1$^{pos}$/2$^{pos}$ CD8 T cells (in 200 μL PBS) were injected in the mouse tail vein. Sample sizes for all experiments were chosen based on previous experiences.

## Mouse CD8 T-cell activation and treatment
TCRP1A CD8 T cells were isolated from the spleen of 7–10 weeks old TCRP1A female mice using anti-mouse CD8α (Ly-2) MicroBeads (Miltenyi Biotech, 130-117-044), and co-cultured with irradiated (10,000 rads) L1210.P1A.B7-1 cells at 1:2 ratio in IMDM (GIBCO, 12440053) complete medium containing β-mercaptoethanol (50 μM, Sigma, M3148). Four days later, alive CD8 T cells were enriched on a Lymphoprep™ gradient (StemCell, 07801, nucleofected (described below) and then cultured in the presence of rhIL-2 (25 U/mL, StemCell, 78036.2) at 21 or 1% O$_2$. For proliferation assay, 10$^6$/mL cells were labeled with Violet CellTrace™ (Invitrogen, C34557) at 5 μM for 20 min at 37 °C according to the manufacturer's instructions, before treatments.

For OT-1 CD8 T-cell activation, splenocytes of 7–12 weeks old CD45.1$^{pos}$ and CD45.2$^{pos}$ OVA-TCR (OT-1) female mice were cultured for 2 days in RPMI (GIBCO 52400041) complete medium containing β-mercaptoethanol (50 μM, Sigma), sodium pyruvate (1 mM, GIBCO 11360070), 1 μg/mL SIINFEKL peptide and rhIL-2 at 25 U/mL. On day 2, cells were washed and seeded in a fresh medium with rhIL-2. After 2 days, alive CD8 T cells were enriched on a Lymphoprep™ gradient (StemCell, 07801) gradient and nucleofected (described below).

## Human CD8 T-cell activation
Peripheral blood mononuclear cells (PBMCs) were derived from the blood of different haemochromatosis patients, courtesy of Saint-Luc University hospital (authorization approval n° CEHF 2021/13SEP/373). The ethical approval encompasses an exemption of informed consent based on the fact that we use only blood that qualifies as "residual human body material", which needs to be collected as a therapeutic measure for patients with haemochromatosis. The option to not opt-out was checked for all patients. CD8 T cells were isolated from PBMCs with CD8 MicroBeads (Miltenyi, 130-045-201) and cultured in the presence of CD3/CD28 beads (Gibco, 11131D) and 25 U/mL rhIL-2 in complete IMDM medium containing 10% human serum. Four days after, alive CD8 T cells were enriched on a Lymphoprep™ gradient (StemCell, 07801 and treated with 20 μM IOX4 (Cayman chemicals, 18181, resuspend in DMSO) in complete IMDM medium in the presence of rhIL-2. This study was approved by the Comité d'Ethique Hospitalo-Facultaire Saint-Luc–UCLouvain under the code CEHF 2021/13SEP/373.

## CD8 T-cell nucleofection
HIF-1α KO, PHD2/3 KO, or double KO CD8 T cells were generated by electroporation of Cas9 ribonucleoprotein complexes as described by ref. 53. Briefly, Alt-R crRNA and Alt-tracrRNA (Integrated DNA Technologies) were mixed in equimolar concentrations and annealed by heating at 95 °C for 5 min and slowly cooled to room temperature. crRNA-tracrRNA duplexes (150 pmol) were incubated with TrueCut Cas9 protein v2 (60 pmol, Thermo Fisher Scientific, A36496) for 10 min to form ribonucleoprotein complexes (RNP). 10–15 × 10$^6$ activated CD8 T cells were resuspended in an 80 μL P4 Primary Cell 4D-Nucleofector X kit (Lonza, V4XP-4024) and incubated with 20 μL RNP for 2 min. Electroporation was performed in the 4D-nucleofector X Unit (Lonza) using the CM137 nucleofector program. After electroporation, cells were cultured in a complete medium supplemented with rhIL-2 and let resting for 4 days. gRNA, crRNA, and tracrRNA were purchased from Integrated DNA technologies (Supplementary Table S1).

## RNA sequencing
HIF-1α KO, PHD2/3 KO, and control TCRP1A CD8 T cells incubated in normoxia or hypoxia for 48 h were collected, washed twice in cold PBS, and RNA was extracted using NucleoSpin RNA kit (Macherey Nagel, 740955) according to the manufacturer's instructions. RNAseq was performed on three biological replicates for each sample by Macrogen Inc. using Illumina NGS workflow. Fastq files were processed using a standard RNAseq pipeline including Trimmomatic-0.38[54] to remove low quality reads, hisat2-2.1.0[55] to align reads to the mouse genome (grcm38), and gene expression levels were evaluated using feature-Counts from subread-2.0.0[56] and Mus_musculus.GRCm38.94.gtf. Differential expression analyses were performed with DESeq2 Bioconductor package v1.38.3[57]. The design used to model the counts was genotype * oxygen_level. Gene set enrichment analyses were done using clusterProfiler v4.6.2[58].

## Flow cytometry staining
For staining of surface markers, cells were resuspended in PBS buffer containing 2% FBS and 2 mM EDTA (FACS buffer) and incubated with CD16/CD32 blocking antibody for 15 min at 4 °C followed by incubation with fluorescent antibodies for 20 min at 4 °C. For staining of intracellular proteins, cells were washed in PBS and resuspended in fixation buffer (eBioscience, 00-5523-00) for 20 min at 4 °C. Samples were then washed twice in permeabilization buffer (eBioscience, 00-5523-00) and incubated with CD16/CD32 blocking antibodies and primary antibodies diluted in permeabilization buffer for 30 min at 4 °C. To detect HIF-1α by flow cytometry analysis, 1 mL of Lyse/fix buffer (BD biosciences, 558049) was added directly into wells containing 200 μL of cells in a medium. After 10 min, samples were collected, centrifuged, and incubated in cold Perm II buffer (BD biosciences, 550852) for 30 min at 4 °C. Cells were then washed in FACS buffer and stained with anti-HIF-1α APC antibody (Cell signaling, D1S7W 1:50).

Tumor single-cell suspensions were obtained by tumor dissociation with the gentleMACS Dissociator (program: m_imp_tumor_01.01 C Tube) and digestion in RMPI medium (Thermo Fisher Scientific, 11875093) containing collagenase I (100 U/mL), collagenase II (100 U/mL), and dispase (1 U/mL) for 15 min at 37 °C. Seventy- and 40-μm-cell strainers were used to eliminate cell aggregates. After red blood cell lysis (Thermo Fisher Scientific, 00-4300-54), cells were resuspended in FACS buffer. An equal number of cells were used for flow cytometry staining. Cells were stained with Fixable Viability dye (15 min, 4 °C) and washed twice before being incubated with CD16/CD32 blocking (15 min, 4 °C) and fluorescent primary antibodies (20 min, 4 °C). For staining of cytosolic proteins, samples were fixed in fixation buffer (eBioscience, 00-5523-00) for 20 min at 4 °C, washed two times in

permeabilization buffer (eBioscience, 00-5523-00), and incubated with CD16/CD32 blocking antibody and fluorescent primary antibodies diluted in permeabilization buffer for 45–60 min at 37 °C.

Samples were acquired at BD LRSFortessa or BD FACSVerse and analyzed with FlowJo 10.7.1. Single stained and Fluorescence Minus One (FMO) samples were used as control. Antibodies and reagents used are listed in Supplementary Table S2.

## Killing assay
Activated scramble, HIF-1α KO or PHD2/3 KO TCRP1A CD8 T cells incubated in normoxia (21% $O_2$) or hypoxia (1% $O_2$) for 72 h were seeded with a mix of fluorescently labeled P1A-expressing P511 (target, labeled with 5 µM CellTrace Violet Cell dye) and P1A-negative P1.204 cells (non-target, labeled with 0.5 µM CellTrace Violet Cell dye) in complete medium. The distinction between target and non-target cells can be achieved by gating on different CellTrace Violet dye intensities by flow cytometry analysis. The cell mix was then cultured for 5 h in normoxia (21% $O_2$) or hypoxia (1% $O_2$), and the proportion of target or non-target cells was analysed by FACS. Specific lysis was calculated using the following formula: $100 \times (1 - \%$ target/% non-target).

## IFN-γ ELISA
After 48 h of incubation under normoxia (21% $O_2$) or hypoxia (1% $O_2$) conditions, 10,000 CD8 T cells were seeded in fresh X-vivo medium (Lonza, BE04-380Q) and co-cultured with 10 000 P511 cells or with P204.1 cells that were pulsed or not with $P1A_{35-43}$ (LPYLGWLVF) antigenic peptide (homemade) under normoxic (21% $O_2$) or hypoxic (1% $O_2$) conditions. After overnight incubation, supernatants were collected, and the amount of IFN-γ present in the medium supernatant was analyzed by ELISA using the DUOSET Mouse IFN-γ (R&D systems, DY485), according to the manufacturer's instructions.

## Quantitative RT-PCR
RNA was extracted using a NucleoSpin RNA kit (Macherey Nagel, 740955) according to the manufacturer's instructions. About 1 µg of RNA was retro-transcribed by the RevertAid RT Kit (Thermo Fisher Scientific, K1691) according to the manufacturer's instructions. qRT-PCR analysis was performed using Takyon Low ROX Probe 5X MasterMix dTTP (Eurogentec, UF-LP5X-C0501) and QuantStudio 3 Real-time PCR instrument (Thermo Fisher Scientific) using the following program: 3 min at 95 °C, then 40 cycles of 10 s at 95 °C and 1 min at 60 °C. Samples were run in technical duplicate. *β-actin* probe and primers were purchased from Eurogentec:

*β-actin*: Fw: 5′-CTCTGGCTCCTAGCACCATGAAG3′, Rv: 5′-GCTGG AAGGTGGACAGTGAG-3′, probe: 5′-FAMATCGGTGGCTCCATCCTGG CTAMRA-3′. Other premade qPCR primer-probe sets were purchased from Integrated DNA Technologies and are listed in Supplementary Table S3. Samples were run in technical duplicates. Data were normalized to *β-actin* expression.

## Western blot
For the detection of cytosolic proteins, cells were lysed in cold RIPA buffer supplemented with a Halt Phosphatase inhibitor cocktail (Thermo Fisher Scientific, 78446) for 20 min on ice. For HIF-1α protein detection, nuclear proteins were extracted using NE-PER nuclear and cytoplasmic extraction reagents (Thermo Fisher Scientific, 78833) according to the manufacturer's instructions. Proteins were quantified by Pierce Protein BCA Assay Kit (Thermo Fisher Scientific, 23225). About 30 µg of total proteins were denatured at 95 °C for 5 min in NuPAGE LDS sample buffer (Thermo Fisher Scientific, NP0007), loaded on NuPAGE 3–8% gels (Thermo Fisher Scientific, EA0378BOX) or 4–12% Bis-Tris gels (Thermo Fisher Scientific, WG1403BOX) and separated by gel electrophoresis using NuPAGE Tris-acetate SDS Running

Buffer (Thermo Fisher Scientific, LA0041), or NuPage MOPS SDS Running Buffer (Thermo Fisher Scientific, NP0001), respectively. Dry transfer was performed by iBlot (Thermo Fisher Scientific) using iBlot Gel Transfer Stacks, nitrocellulose (Thermo Fisher Scientific, IB23002), and a transfer program P0 (1 min at 20 V, 4 min at 23 V, 2 min at 25 V). After blocking in TBS 0.1% Tween 20 (TBS-T) with 5% nonfat dry milk (Nestlé) for 1 h at room temperature, the membranes were incubated overnight at 4 °C with primary antibody anti-HIF-1α (Cayman chemicals, 10006421, 1:3000), anti-granzyme B (Abcam, ab255598, 1:1000), anti-perforin (Cell signaling, 31647, 1:1000), or anti-HIF-2α (Novus Biological, NB100-122, 1:1000). After washing in TBS-T, membranes were incubated with HRP-linked anti-rabbit secondary antibody (Cell signaling, cl7074S, 1:5000) in TBS-T containing 5% nonfat dry milk (Nestlé) for 1 h at room temperature. Alternatively, membranes were incubated with HRP-linked antibodies: HRP-anti-HDAC1 (Cell signaling, 59581, 1:5000), HRP-anti-GAPDH (Cell signaling, 8884S, 1:3000) or HRP-b-tubulin (Cell signaling, 5346S, 1:3000) for 1 h at room temperature. HRP signal was revealed using SuperSignal West Pico or Femto (Thermo Fisher Scientific, 34580- 34095) and detected with Fusion Solo S (Vilber Lourmat).

## Tumor rechallenging experiment
Concurrent tumor implantation and adoptive cell transfer were conducted by subcutaneously injecting mice with $10^6$ MC38-OVA cells and intravenously injecting them with $3 \times 10^6$ scramble or PHD2/3 KO CD8 T cells. Tumor formation was monitored for 4 weeks, and mice that developed tumors were recorded. After four weeks, mice without tumors were rechallenged with $10^6$ MC38-OVA tumor cells, and tumor formation was subsequently monitored. For analysing the re-expansion potential of transferred T cells, concurrent tumor implantation and adoptive cell transfer involved subcutaneously injecting mice with $10^6$ MC38-OVA cells while simultaneously administering an intravenous injection of $3 \times 10^6$ scramble or PHD2/3 KO CD8 T cells. After 4 weeks, mice without tumors from each group were intravenously injected with $10^6$ MC38-OVA tumor cells. Subsequently, blood, lymph nodes, and spleen samples were collected 24 h later and analysed for the presence of CD45.1 positive cells.

## Seahorse analysis
A Seahorse XFe96 Bioanalyser (Agilent) was used to determine basal OCR and ECAR of nucleofected normoxic or hypoxic TCRP1A CD8 T cell based on a protocol described by ref. 59. Briefly, CD8 T cells were washed in assay medium (IMDM medium, HEPES-free, phenol red-free, 50 U/mL penicillin and 50 mg/mL, L-arginine (0.55 mM), L-asparagine (0.24 mM), glutamine (1.5 mM), rhIL-2 (25 U/ml)) before being plated onto Seahorse cell culture plates coated with poly-L-lysine (Corning) at $2 \times 10^5$ cells per well. After adherence and equilibration, basal OCR and ECAR were measured on Seahorse XFe96 Bioanalyser (Agilent).

## Glucose uptake
To analyze glucose uptake in CD8 T cells, luminescence-based assay Glucose uptake-Glo (Promega, J1341) was performed according to the manufacturer's instructions. Briefly, $1 \times 10^5$ CD8 T cells were washed in PBS and incubated with 1 mM of 2-deoxyglucose for 10 min at 37 °C. Immediately after, stop buffer, neutralization buffer, and detection reagent were added per the manufacturer's instructions. Luminescence was read with a GloMax microplate reader (Promega).

## Lactate measurements
Secreted lactate by CD8 T cells was measured using a luminescence-based assay Lactate-Glo (Promega, J5021) used as per manufacturer's instructions. Briefly, $3 \times 10^5$ human or $1.5 \times 10^5$ mouse CD8 T cells were

seeded in 800 μL of complete medium and incubated for 48 h at 37 °C under normoxic or hypoxic conditions. Supernatants were diluted 1:100 in PBS before being used for lactate quantification.

## Statistics and reproducibility

Data entry and analyses were performed in a blinded fashion. Randomization was performed for in vivo experiments. Outliers were excluded from the analyses. Statistical analyses were performed with GraphPad Prism software (Version 9.2) using one-way or two-way ANOVA with Tukey's correction for multiple comparisons; survival curves were compared with log-rank (Mantel–Cox) test. In the figures, asterisks denote statistical significance (****$p < 0.0001$).

## Reporting summary

Further information on research design is available in the Nature Portfolio Reporting Summary linked to this article.

## Data availability

All data supporting the findings of this study are available within the Article and its Supplementary Information. The raw and processed RNA sequencing data generated in this study have been deposited in the Gene Expression Omnibus (GEO) database under accession code GSE271367 (https://www.ncbi.nlm.nih.gov/geo/query/acc.cgi?acc=GSE271367. All raw data generated in this study are provided in the Supplementary Information/Source Data file. Source data are provided with this paper.

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

## Acknowledgements

We thank Pedro Gomez and the Platform Laboratory Animal Facility for mouse colony production, Cyril Corbet for help with Seahorse analysis, Luc Pilotte for technical support, and Isabelle Grisse for editorial assistance. T.D. was supported by FNRS-Télévie (7.4597.18). V.F. was supported by FNRS-Télévie (7. 4501.15F), FRS-FNRS- chargé de recherches (FC-7929), Foundation against cancer Fundamental Mandate (FFC, 2019-092). J.Z. was supported by Fondation Contre le Cancer (grant no. 2019-094). This work was supported by Walloon Excellence in Life Sciences (WELBIO-CR-2015A-07, WELBIO-CR-2019C-05), WALInnov grant from the Walloon region (IMMUCAN, 1610119), Fonds pour la Recherche Scientifique (FNRS), Belgium (grant nos EOS O000518F and PDR T.0091.18); Fondation contre le Cancer, Belgium (grant no. 2018-090); Ludwig Cancer Research; and de Duve Institute.

## Author contributions

T.D.: study design, development of methodology, data acquisition, data analysis and interpretation, writing; V.F.: study design and supervision, data analysis and interpretation; M.F. and L.B.: development of methodology, data acquisition, and analysis; A.L.: RNAseq data analysis; J.Z.: supervision, writing; B.J.V.d.E. conception, supervision, and funding acquisition.

## Competing interests

The authors declare no competing interests.
