## [Peer Review File · Nature Communications]

Enhanced tumour response to adoptive T cell therapy with PHD2/3-deficient CD8 T cellsREVIEWER COMMENTS

Reviewer #1 (Remarks to the Author):

Dvorakova et al demonstrate that genetic engineering of isolated T cells to enforce HIF1a activation leads to enhanced anti-tumor activity. The work is well done, and experiments are elegantly designed. However, stabilization of HIF1a in CD8+ T cells had already been shown to increase T cell function in various settings including anti-tumor activity (Doedens, Nature Immunology, 2014). Moreover, it had already been shown that HIF1a/hypoxia increases the activating receptor CD137 or 41BB (Palazon, Cancer Discovery, 2012). The major difference that this current manuscript brings is the fact that activation of HIF1a in T cells is induced after T cell development and after T cell activation, bypassing any developmental effects of HIF1a. Most of the findings are very similar to the conditional knockout mice in previous publications, which validates previous work but reduces originality. The fact that HIF1a is enforced after T cell activation rather than constitutively in knockout mice should have been better exploited to bring novelty to the work and discover something new beyond what had already been published. There are a few points that if addressed can expand the potential therapeutic use of this axis that can distinguish this work from the previous literature.

Major points

1. In previous work where VHL^{-/-} CD8 T cells were analyzed, this was done with Lck-CRE x Vhl^{-flox/flox} mice. In this case, naïve T cells had already activation of HIF1a when they were activated. Here, instead, deletion of PHD2/3 is done after activation, which is suggested in the Introduction as a possible avenue for CAR T cell therapy and other T cell-based therapies. A side-by-side comparison of electroporation of PHD2/3 of naïve T cells and then activation with electroporation of already activated T cells could inform about the advantage of the approach used here. It can also reveal some new biological insights beyond the published work. Is there any difference in any of the exhaustion or activation markers when HIF1a is induced before or after T cell activation?
2. In line with the question above, activation of HIF1a after T cell activation may lead to differences in their long-term survival. The fate decision towards an Effector phenotype in detriment of a "Memory" or "Stem-like" or "Precursor exhausted" phenotype may lead to acute response but subsequent loss of the T cells. A broad body of literature both in endogenous anti-tumor CD8 T cells and in CAR T cells is indicating that long-lived memory T cells are essential for responses in patients whereas short-lived effector T cells are quickly lost. Quantifying TCF1, and Slamf6 by flow cytometry would inform of the % of precursor cells once HIF1a is activated. In line with this, if engineering PHD2 is suggested as a therapeutic avenue, the time that T cells survive and are retained should be measured. Can the mice be re-challenged with tumors and see whether CD45.1 WT or PHD2/3^{-/-} T cells expand? This may explain differences in expression of Ifng compared to conditional KO mice.
3. Glycolysis is suggested as the mechanism by which HIF1a increases T cell function. A double KO with a gene in the glycolytic pathway or the transporter GLUT1 would prove this point. With the presented data, although tantalizing, it is only a correlation.
4. The last figure uses an inhibitor of PHD2/3 for treatment in vitro of human T cells. If this is intended for future possible therapies, this should be tested in NSG mice with human T cells and if not possible due to systemic toxicity, this should be discussed and clarified that its usage for therapy will not be possible.
5. Flow cytometry plots and gating strategies are key for evaluation of the work and reproducibility. No flow cytometry plot is shown in the entire paper, only barr graphs

Minor

Fig3a schematic is confusing: it says "1M tumor cells (CD45.2)" which makes it look like the tumor cells express CD45.2. It also says that the adoptively transferred OT1 T cells are CD45.1/2 but they are only CD45.1 while the recipient mice are CD45.2. Or are they homozygous CD45.1 and CD45.2 In any case, they are visualized by CD45.1.

Reviewer #2 (Remarks to the Author):

General comments:

The paper by Dvorakova demonstrates that loss of PHD2/3 in CD8+ T cells promotes enhanced antitumor immunity. The precise is that PHD2/3 loss stabilizes HIF-1alpha resulting in a shift to glycolysis and boost in T cell responses. This conclusion is based on using various preclinical tumor models and ex vivo assessment of PHD2/3 KO T cells.

My general comment is that this is an excellent manuscript. The knockout model and systems are well-designed and executed. All the data are clearly presented and support the model / conclusions of the paper. Having said that, it is not clear how the main contention that HIF-1 is responsible for regulating T cell activity, differs from the original study by Doedens et al., (and follow-up studies). While I recognize that VHL versus PHD2/3 are different targets, the final outcome of manipulating these two pathways is the same. Therefore, how do the authors feel that their work brings significantly new insight into this pathway on T cell regulation of antitumor immunity?

There are also a few items I would like to see addressed:

Fig 1 – The phenotype is striking and convincing. There are no assessments of the T cells after ACT. This is crucial and additional experiments are needed on isolated PHD2/3 CD8+ T cell to show enhanced function and immunological consequences of PHD2/3 KO (e.g. ex vivo cytokine secretion (ELISPOT, ICS, etc), metabolic changes (seahorse), flow cytometry T cell differentiation markers, etc).

Fig 2f – the improvement in cytolytic activity between PHD2/3 KO vs HIF-1 KO is not equivalent regardless of oxygen tension. PHD2/3 KO is universally better at improving CTL activity. This seems to be contrary to the authors suggestion that the effects of PHD2/3 KO is specifically through stabilization of HIF-1a. Can the authors comment on this point?

Some additional controls need to show how PHD2/3 affects specificity: 1) Expression of HIF-2alpha and 2) targets of HIF-1alpha.

RESPONSE TO REFEREES

Reviewer #1 (Remarks to the Author):

Dvorakova et al demonstrate that genetic engineering of isolated T cells to enforce HIF1a activation leads to enhanced anti-tumor activity. The work is well done, and experiments are elegantly designed. However, stabilization of HIF1a in CD8+ T cells had already been shown to increase T cell function in various settings including anti-tumor activity (Doedens, Nature Immunology, 2014). Moreover, it had already been shown that HIF1a/hypoxia increases the activating receptor CD137 or 41BB (Palazon, Cancer Discovery, 2012). The major difference that this current manuscript brings is the fact that activation of HIF1a in T cells is induced after T cell development and after T cell activation, bypassing any developmental effects of HIF1a. Most of the findings are very similar to the conditional knockout mice in previous publications, which validates previous work but reduces originality. The fact that HIF1a is enforced after T cell activation rather than constitutively in knockout mice should have been better exploited to bring novelty to the work and discover something new beyond what had already been published. There are a few points that if addressed can expand the potential therapeutic use of this axis that can distinguish this work from the previous literature.

Author response:

We would like to thank the reviewer for this valuable comment. As pointed by the reviewer, much of the existing literature relies on studies utilizing knockout mice, wherein the impact on T-cell development and differentiation is undeniable. We aimed to test a strategy that is closer to the clinical situation. However, translating such findings into viable human treatments presents significant challenges. While a general inhibition of PHD2/3 could theoretically offer a solution, such inhibition may lead to the upregulation of HIF-1 α in all cell types, including tumor cells, a factor often associated with tumor growth, and in other immune cells, such as macrophages, in which HIF-1 α was shown to increase the T-cell suppressive properties.

In light of these complexities, our study proposes another approach: specifically stabilizing HIF-1 α in CD8 T cells, particularly in the setting of adoptive cell therapy with CAR-T cells or TCR-T cells therapy or TIL therapy. Our findings not only underscore but also substantiate the notion that deleting PHD2/3 in adoptively transferred T cells can enhance their cytotoxicity and persistence within tumor environments. Interestingly, the new experiments we did upon the reviewer's suggestion (see below) addressing the impact of PHD2/3 inhibition on naïve T cells prior to activation revealed an unexpected inhibition of T-cell activation. This observation raises relevant questions regarding the non-selective use of PHD2/3 inhibitors in cancer treatment. It underscores the critical importance of our proposed strategy.

In essence, our research aims for a more refined and selective approach to modulating PHD2/3 activity in T-cell-based cancer therapies. By focusing on the precise manipulation of these enzymes within T cells, we aim to maximize therapeutic efficacy while minimizing potential negative effects.

As suggested by the reviewer, we have added a paragraph highlighting these important points in the manuscript, from line 83 to 101, as follows:

“Several studies involving CD8 T cells isolated from conditional knockout (KO) mice for HIF, PHD or VHL have demonstrated the potential of stabilizing HIF to enhance T-cell function, offering exciting prospects for therapeutic strategies.^{9,10,19} While these findings are encouraging, it is crucial to acknowledge that these studies relied on CD8 T cells sourced from PHD-KO or VHL-KO mouse strains, in which modifications in HIF signaling occur during the whole process of lymphocyte development and differentiation. It remains to be determined whether HIF stabilization also increases effector functions in already differentiated T cells. Moreover, translating such findings into human therapy presents significant challenges. One could think of

using pharmacological inhibitors of PHD2/3, but these would have the drawback of upregulating HIF-1 α in all cells, including tumor cells, thereby promoting tumor growth^{20,21}. It would also stabilize HIF-1 α in other immune cells, potentially impairing anti-tumor immunity. Indeed, expression of HIF-1 α in macrophages was shown to increase their ability to suppress T-cell function and proliferation²². To navigate these complexities, targeting HIF-1 α stabilization specifically in CD8 T cells, especially in conjunction with CAR T or TIL therapy holds promise, where T cells are typically engineered after differentiation and activation.²³ However, in contrast to previously established findings that suggested HIF-1 α stabilization in VHL-KO or PHD-KO mouse models enhances T-cell responses^{10,19}, the effect of HIF-1 α stabilization after CD8 T cell activation on their antitumor responses remains uncertain. This uncertainty arises from findings such as those reported by Zhang et al., who observed a delayed growth of B16-OVA tumors in mice who received adoptive transfers of HIF-1 α knockdown ovalbumin-specific (OT-1) CD8 T cells.²⁴

Major points

1. In previous work where VHL-/- CD8 T cells were analyzed, this was done with Lck-CRE x Vhl-flox/flox mice. In this case, naïve T cells had already activation of HIF1a when they were activated. Here, instead, deletion of PHD2/3 is done after activation, which is suggested in the Introduction as a possible avenue for CAR T cell therapy and other T cell-based therapies. A side-by-side comparison of electroporation of PHD2/3 of naïve T cells and then activation with electroporation of already activated T cells could inform about the advantage of the approach used here. It can also reveal some new biological insights beyond the published work. Is there any difference in any of the exhaustion or activation markers when HIF1a is induced before or after T cell activation?

Author response:

We would like to express our appreciation to the reviewer for raising this insightful point. Indeed, a comparison between T cell deletion for PHD2/3 KO in both naïve and activated T cells would offer valuable insights. However, our lab currently does not have access to PHD2 KO mice, and our attempts to reach out to researchers who have previously utilized them have not yielded a solution.

Nevertheless, we compared the effect PHD2 deletion in naïve T cells before activation and in those already activated. Ideally, we would have preferred to knock out PHD2 in naïve T cells and directly compare its impact on exhaustion or activation markers. However, we encountered technical challenges in achieving this. Primarily, naïve T cells proved more challenging to engineer, whether through electroporation or virus transduction, resulting in insufficient KO cell yield. Additionally, the survival of naïve T cells requires immediate activation, which does not fit with the time required for genetic modification.

Consequently, we opted to utilize the PHD2 inhibitor IOX4 to treat both naïve and activated T cells and assess activation markers. IOX4 competitively binds to and displaces 2-oxoglutarate (2OG) at the active site of PHD2, leading to immediate inhibition of PHD2 activity. In our experiments, we treated naïve T cells with IOX4 prior to activation and also treated activated T cells four days post-activation with IOX4. Flow cytometry analysis was conducted seven days post-activation for naïve T cells and three days post-activation for activated T cells.

As expected, for activated T cells, IOX4 treatment resulted in increased expression of Granzyme B and Lag3 compared to the control group under normoxic conditions. Surprisingly, in naïve T cells treated with IOX4, we observed decreased expression of both Granzyme B and Lag3.

These new findings have been incorporated into Supplementary Figure 3E, and the corresponding text has been updated in the manuscript between line 182 and line 189, as follows:

“To deepen our understanding of the impact of PHD2/3 inhibition in T cells before and after T cell activation, we subjected both naïve and activated CD8 T cells to IOX4, a specific PHD2 inhibitor, and evaluated its effects on the expression of various functional markers. As anticipated, in activated T cells, IOX4 treatment led to elevated expression of Granzyme B, Tox and Lag3 compared to the control group under normoxic conditions (Suppl Fig 3E). However, unexpectedly, in naïve T cells treated with IOX4, we observed reduced expression of both Granzyme B, Tox and Lag3, suggesting that PHD2 inhibition may have distinct consequences on T-cell function depending on their activation status (Suppl Fig 3E). These findings supported our approach to inactivate PHD2/3 in activated T cells before ACT.”

Supplementary Figure 3E: Flow cytometry analysis of the expression levels of Granzyme B, Tox, and Lag3 on naïve or activated CD8 T cells, with or without treatment with 10 μM IOX4. Naïve T cells were treated with IOX4 for 3 hours followed by immediate activation, with FACS analysis performed 7 days after activation. Activated T cells were treated with IOX4 4 days after activation, with FACS analysis performed 7 days after activation.

2. In line with the question above, activation of HIF1a after T cell activation may lead to differences in their long-term survival. The fate decision towards an Effector phenotype in detriment of a “Memory” or “Stem-like” or “Precursor exhausted” phenotype may lead to acute response but subsequent loss of the T cells. A broad body of literature both in endogenous anti-tumor CD8 T cells and in CAR T cells is indicating that long-lived memory T cells are essential for responses in patients whereas short-lived effector T cells are quickly lost. Quantifying TCF1, and Slamf6 by flow cytometry would inform of the % of precursor cells once HIF1a is activated. In line with this, if engineering PHD2 is suggested as a therapeutic avenue, the time that T cells survive and are retained should be measured. Can the mice be re-challenged with tumors and see whether CD45.1 WT or PHD2/3-/- T cells expand? This may explain differences in expression of Ifng compared to conditional KO mice.

Author response:

We would like to thank the reviewer for this valuable suggestion. As suggested, we evaluated the expression of markers indicative of “Stem-like” or “precursor” T cell phenotypes in MC38-OVA tumor-bearing mice following adoptive cell transfer of either scramble or PHD2/3 KO T cells. As depicted in Supplementary Figure 6 and detailed below, our findings revealed that mice receiving ACT with PHD2/3 KO T cells exhibited an increased percentage and expression of Slamf6 and TCF1. This indicates that PHD2/3 knockout enhances the stem-like characteristics of transferred T cells, potentially contributing to improved persistence and antitumor efficacy.

Additionally, as suggested, we tried to assess and compare T cell persistence within the tumor microenvironment of mice treated with either scramble or PHD2/3 KO T cells. We did so through rechallenge experiments in which we monitored tumor rejection and T-cell expansion.

For the rechallenge experiment, given that no mice experienced complete tumor rejection in the control group and only a minimal number in the PHD2/3 KO group, we modified the experimental setup. Concurrent tumor implantation and adoptive cell transfer were performed to preserve T-cell activation by tumor cells, while ensuring improved tumor rejection. As outlined in the table below and depicted in Supplementary Figure 6B, approximately 40% of mice treated with scramble T cells developed tumors, versus none in the PHD2/3 KO group. We subsequently rechallenged tumor-free mice with identical tumor cells after a 4-week interval. As shown in the table below and in Figure 6B, we observed a tumor recurrence rate of approximately 33% in the scramble group compared to a mere 5% in the PHD2/3 KO group.

Finally, in parallel, we conducted a rechallenging experiment involving the intravenous injection of 1 million MC38-OVA tumor cells. Subsequently, blood and lymph node samples were collected one day post-tumor injection to assess the expansion of CD45.1 cells. Notably, upon analyzing the percentage of CD45.1 cells, as illustrated below and in Supplementary Figure 6C, a marked increase was observed in the lymph nodes of the PHD2/3 group compared to the control group.

These data suggest that PHD2/3 KO cells have the feature of long-lived memory cells and display better long-term persistence in vivo.

These new findings have been incorporated into Supplementary Figure 6, and the corresponding text has been updated in the manuscript between line 249 and line 265, as follows:

“Further investigation into stem-like markers Slamf6 and TCF1 expression on transferred T cells revealed a notable difference between PHD2/3 KO and Scramble CD8+ T cells. Specifically, there was a higher percentage of Slamf6-positive CD8+ T cells among the transferred T cells in mice that received PHD2/3 KO T cells compared to those that received Scramble T cells (Supplementary Figure 6A). A similar pattern was observed for TCF1, another marker widely used to define “stem-like” or precursor T cells. In line with these phenotypic changes, rechallenge experiments demonstrated better long-term protective effect of PHD2/3 KO OT-1 T cells as compared to control OT-1 T cells (Suppl Fig 6B). In these experiments, in which simultaneous tumor implantation and adoptive cell transfer were conducted, approximately 40% of mice treated with Scramble T cells developed tumors, vs none in the PHD2/3 KO group. We then subjected tumor-free mice from both groups to an identical tumor cell rechallenge 4 weeks later. The Scramble group demonstrated a tumor recurrence rate of approximately 33%, whereas the PHD2/3 KO group exhibited a mere 5% recurrence rate (Suppl Fig 6B). Furthermore, upon rechallenging mice with intravenous injections of MC38-OVA tumor cells, a notable increase in the expansion of transferred CD8 T cells was observed in the group that received adoptive cell therapy (ACT) of PHD2/3 KO OT-1 T cells compared to the control group (Suppl Fig 6C). This finding provides additional confirmation of the superior “memory” or “progenitor” phenotype exhibited by the PHD2/3 KO T cells (Suppl Fig 6A).”

Supplementary Figure 6

Supplementary Figure 6: Impact of PHD2/3 deletion in CD8 T cells on their memory phenotype.

(A) Flow cytometry analysis was performed to assess the *Slamf6pos* and *Tim3pos* populations as well as the *Slamf6neg* and *Tim3pos* populations among CD45.1 positive CD8 T cells within the tumors of mice that received adoptive cell transfer of either Scramble or PHD2/3 KO OT-1 CD8 T cells.

(B) Concurrent tumor implantation and adoptive cell transfer were conducted by subcutaneously injecting mice with 1 million MC38-OVA cells and intravenously injecting them with 3 million Scramble or PHD2/3 KO CD8 T cells. Tumor formation was monitored for 4 weeks, and mice that developed tumors were recorded. After 4 weeks, mice without tumors were rechallenged subcutaneously with 1 million MC38-OVA tumor cells, and tumor formation was subsequently monitored.

(C) Analysis of the expansion of transferred Scramble or PHD2/3 KO CD8 T cells upon tumor cell rechallenge. Concurrent tumor implantation and adoptive cell transfer involved subcutaneously injecting mice with 1 million MC38-OVA cells while simultaneously administering an intravenous injection of 3 million Scramble or PHD2/3 KO CD8 T cells. After 4 weeks, mice without tumors from each group were intravenously injected with 1 million MC38-OVA tumor cells. Subsequently, blood, lymph nodes, and spleen samples were collected 24 hours later and analysed for the presence of CD45.1 positive cells.

Data in A and C are mean \pm SEM. *p* values are calculated by one-way ANOVA with Tukey's multiple comparison test.

3. Glycolysis is suggested as the mechanism by which HIF1a increases T cell function. A double KO with a gene in the glycolytic pathway or the transporter GLUT1a would prove this point. With the presented data, although tantalizing, it is only a correlation.

Author response:

We thank the reviewer for this excellent suggestion. We used 2-Deoxyglucose (2DG) to inhibit glycolysis. This was technically easier than another KO, and potentially more complete than targeting only GLUT1, given that glucose entry into cells is mediated by various glucose transporters, including GLUT1 and GLUT3. 2-Deoxyglucose (2-DG), a non-metabolizable glucose analog, enters cells and accumulates, thereby blocking hexokinase 2.

As depicted below and detailed in Fig 4J, the inhibition of glycolysis effectively abolished the enhanced T-cell function induced with PHD2/3 KO. This underscores the crucial role of glucose metabolism in mediating the enhanced T cell function associated with PHD2/3 deficiency. These new findings have been now included in Figure 4J, and the corresponding text has been updated in the manuscript between line 313 and line 316, as follows:

“The involvement of glycolysis in the increased effector functions of PHD2/3 KO CD8 T cells was supported by the observation that inhibiting glycolysis using 2-deoxy-D-glucose (2-DG) abolished the increased Granzyme B expression induced by PHD2/3 KO (Figure 4J).”

Figure 4J: Flow cytometry analysis of Granzyme B expression on scramble or PHD2/3 KO OT-1 CD8 T cells treated or not with 5 mM 2-deoxy-D-glucose (2-DG).

4. The last figure uses an inhibitor of PHD2/3 for treatment in vitro of human T cells. If this is intended for future possible therapies, this should be tested in NSG mice with human T cells and if not possible due to systemic toxicity, this should be discussed and clarified that its usage for therapy will not be possible.

Author response:

We are not proposing systemic treatment with PHD inhibitors, which would lead to the upregulation of HIF-1 α in all cell types, including tumor cells, a factor often associated with tumor growth, and in other immune cells, such as macrophages, in which HIF1a was shown to increase the T-cell suppressive properties.

The whole idea of our study is to target PHD only in T cells. Despite numerous attempts we have not been able to KO PHD2/3 in human CD8 T cells to be used in humanized NSG models.

We have clarified these points in the manuscript at several places, as follows:

Introduction, lines 89-94

“Moreover, translating such findings into human therapy presents significant challenges. One could think of using pharmacological inhibitors of PHD2/3, but these would have the drawback of upregulating HIF-1 α in all cells, including tumor cells, thereby promoting tumor growth^{20,21}. It would also stabilize HIF-1 α in other immune cells, potentially impairing anti-tumor immunity. Indeed, expression of HIF-1 α in macrophages was shown to increase their ability to suppress T-cell function and proliferation²².”

Results, lines 327-332

“Even though we do not envision systemic treatment with PHD inhibitors due to opposite effects on other cells, these data validate the notion that HIF-1 α stabilization also increased effector functions in human CD8 T cells. Collectively, these findings mirror our observations with mouse CD8 T cells and suggest that the stabilization of HIF-1 α in T cells before ACT could offer a promising strategy to sustain T-cell effector functions, potentially leading to improved clinical outcomes of T-cell therapies.”

Discussion, lines 391-402

“While our findings suggest promising prospects for leveraging PHD2/3 to enhance T cell therapy, it is crucial to consider the broader implications of PHD2/3 inhibition. A systemic inhibition of PHD2/3 with pharmacological agents may exert unintended effects on other cell types due to the ubiquitous expression of the PHD/HIF system. Of particular concern is the role of HIF-1 α in tumor progression and metastasis, which requires cautious consideration in therapeutic interventions targeting the PHD2/3- HIF-1 α axis^{20,21,51}. Another concern is the effect on other immune cells, such as macrophages, in which an HIF-1 α -mediated hypoxia response was shown to increased T-cell suppression²². In response to these concerns, our proposed strategy focuses on selectively inhibiting PHD2/3 activity within CD8 T cells in the context of ACT, thereby avoiding potential off-target effects while enhancing T-cell function. This targeted approach holds promise for optimizing the efficacy and safety of T cell-based immunotherapies and warrants further investigation in preclinical and clinical settings.”

5. Flow cytometry plots and gating strategies are key for evaluation of the work and reproducibility. No flow cytometry plot is shown in the entire paper, only barr graphs

Author response:

We would like to thank the reviewer for the comment and the gating strategy is now added in Supplementary Fig 3D, 4B, 6A and 6C.

Minor

Fig3a schematic is confusing: it says “1M tumor cells (CD45.2)” which makes it look like the tumor cells express CD45.2. It also says that the adoptively transferred OT1 T cells are CD45.1/2 but they are only CD45.1 while the recipient mice are CD45.2. Or are they homozygous CD45.1 and CD45.2 In any case, they are visualized by CD45.1.

Author response:

Following the reviewer's suggestion, we have modified the text in the figure legend to clarify this point. The modified figure legend can be found below:

"Schematic representation of the experimental design. CD45.2 C57/BL6 mice bearing LLC-OVA or MC38-OVA tumors were treated with CD45.1^{pos} CD45.2^{pos} OT-1 CD8 T cells. CD45.1 was used as a marker to differentiate between adoptively transferred (CD45.1^{pos} CD45.2^{pos}) and endogenous tumor infiltrating CD8 T cells (CD45.1^{neg} CD45.2^{pos})."

Reviewer #2 (Remarks to the Author):

General comments:

The paper by Dvorakova demonstrates that loss of PHD2/3 in CD8+ T cells promotes enhanced antitumor immunity. The precise is that PHD2/3 loss stabilizes HIF-1alpha resulting in a shift to glycolysis and boost in T cell responses. This conclusion is based on using various preclinical tumor models and ex vivo assessment of PHD2/3 KO T cells.

My general comment is that this is an excellent manuscript. The knockout model and systems are well-designed and executed. All the data are clearly presented and support the model / conclusions of the paper. Having said that, it is not clear how the main contention that HIF-1 is responsible for regulating T cell activity, differs from the original study by Doedens et al., (and follow-up studies). While I recognize that VHL versus PHD2/3 are different targets, the final outcome of manipulating these two pathways is the same. Therefore, how do the authors feel that their work brings significantly new insight into this pathway on T cell regulation of antitumor immunity?

Author response:

We would like to thank the reviewer for this valuable comment.

As pointed by the reviewer, much of the existing literature relies on studies utilizing knockout mice, wherein the impact on T-cell development and differentiation is undeniable. We aimed to test a strategy that is closer to the clinical situation. However, translating such findings into viable human treatments presents significant challenges. While a general inhibition of PHD2/3 could theoretically offer a solution, such inhibition may lead to the upregulation of HIF-1 α in all cell types, including tumor cells, a factor often associated with tumor growth, and also in other immune cells, such as macrophages, in which HIF1a was shown to increase the T-cell suppressive properties.

In light of these complexities, our study proposes another approach: specifically stabilizing HIF-1 α in CD8 T cells, particularly in the setting of adoptive cell therapy with CAR-T cells or TCR-T cells therapy or TIL therapy. Our findings not only underscore but also substantiate the notion that deleting PHD2/3 in adoptively transferred T cells can enhance their cytotoxicity and persistence within tumor environments. Interestingly, the new experiments we did upon the suggestion of reviewer 1 addressing the impact of PHD2/3 inhibition on naïve T cells prior to activation revealed an unexpected inhibition of T-cell activation. This observation raises relevant questions regarding the non-selective use of PHD2/3 inhibitors in cancer treatment. It underscores the critical importance of our proposed strategy.

In essence, our research aims for a more refined and selective approach to modulating PHD2/3 activity in T-cell based cancer therapies. By focusing on the precise manipulation of these enzymes within T cells, we aim to maximize therapeutic efficacy while minimizing potential negative effects.

As suggested by the reviewer, we have added a paragraph highlighting these important points in the manuscript, from line 83 to 101, as follows:

“Several studies involving CD8 T cells isolated from conditional knockout (KO) mice for HIF, PHD or VHL have demonstrated the potential of stabilizing HIF to enhance T-cell function, offering exciting prospects for therapeutic strategies.^{9,10,19} While these findings are encouraging, it is crucial to acknowledge that these studies relied on CD8 T cells sourced from PHD-KO or VHL-KO mouse strains, in which modifications in HIF signaling occur during the whole process of lymphocyte development and differentiation. It remains to be determined whether HIF stabilization also increases effector functions in already differentiated T cells. Moreover, translating such findings into human therapy presents significant challenges. One could think of using pharmacological inhibitors of PHD2/3, but these would have the drawback of upregulating HIF-1 α in all cells, including tumor cells, thereby promoting tumor growth^{20,21}. It would also stabilize HIF-1 α in other immune cells, potentially impairing anti-tumor immunity. Indeed, expression of HIF-1 α in macrophages was shown to increase their ability to suppress T-cell function and proliferation²². To navigate these complexities, targeting HIF-1 α stabilization specifically in CD8 T cells, especially in conjunction with CAR T or TIL therapy holds promise, where T cells are typically engineered after differentiation and activation.²³ However, in contrast to previously established findings that suggested HIF-1 α stabilization in VHL-KO or PHD-KO mouse models enhances T-cell responses^{10,19}, the effect of HIF-1 α stabilization after CD8 T cell activation on their antitumor responses remains uncertain. This uncertainty arises from findings such as those reported by Zhang et al., who observed a delayed growth of B16-OVA tumors in mice who received adoptive transfers of HIF-1 α knockdown ovalbumin-specific (OT-1) CD8 T cells.²⁴”

There are also a few items I would like to see addressed:

1. Fig 1 – The phenotype is striking and convincing. There are no assessments of the T cells after ACT. This is crucial and additional experiments are needed on isolated PHD2/3 CD8+ T cell to show enhanced function and immunological consequences of PHD2/3 KO (e.g. ex vivo cytokine secretion (ELISPOT, ICS, etc), metabolic changes (seahorse), flow cytometry T cell differentiation markers, etc).

Author response:

We thank the reviewer for these suggestions. Regarding Figure 1, our study was conducted in a spontaneous model where tumor formation typically requires months. However, due to variations in tumor growth rates, it took even longer to accumulate a sufficient number of mice for the study. Given the importance of investigating tumor growth, we prioritized this aspect of the research. Because we focused on tumor growth, we did not analyze T cells post-adoptive cell transfer (ACT) in this particular model. Instead, we conducted T-cell analysis after ACT in other models, specifically the MC38-OVA and LLC-OVA models. In Suppl Fig 6A, we observed enhanced T cell persistence in mice receiving T cells deficient in PHD2/3 compared to control T cells. Intracellular staining (ICS) analysis revealed increased expression of Granzyme B and Perforin in PHD2/3 knockout (KO) T cells post-transfer, indicating enhanced T cell killing function.

In addition, following the reviewer's suggestion and as suggested also by reviewer 1, we evaluated the expression of markers indicative of "Stem-like" or "precursor" T cell phenotypes in MC38-OVA tumor-

bearing mice following adoptive cell transfer of either scramble or PHD2/3 KO T cells. As depicted in Supplementary Figure 6 and detailed below, our findings revealed that mice receiving ACT with PHD2/3 KO T cells exhibited an increased percentage and expression of Slamf6 and TCF1. This indicates that PHD2/3 knockout enhances the stem-like characteristics of transferred T cells, potentially contributing to improved persistence and antitumor efficacy.

In terms of metabolic changes, we acknowledge the potential benefits of conducting Seahorse analysis using isolated CD45.1 CD8 T cells. However, we are currently encountering technical obstacles. Seahorse analysis demands a considerable number of T cells to ensure reproducibility of results. Regrettably, despite our efforts in FACS sorting, we have been unable to obtain a sufficient quantity of T cells for this purpose.

These new findings have been incorporated into Supplementary Figure 5, and the corresponding text has been updated in the manuscript between line 249 and line 265, as follows:

“Further investigation into stem-like markers Slamf6 and TCF1 expression on transferred T cells revealed a notable difference between PHD2/3 KO and Scramble CD8+ T cells. Specifically, there was a higher percentage of Slamf6-positive CD8+ T cells among the transferred T cells in mice that received PHD2/3 KO T cells compared to those that received Scramble T cells (Supplementary Figure 6A). A similar pattern was observed for TCF1, another marker widely used to define “stem-like” or precursor T cells. In line with these phenotypic changes, rechallenge experiments demonstrated better long-term protective effect of PHD2/3 KO OT-1 T cells as compared to control OT-1 T cells (Suppl Fig 6B). In these experiments, in which simultaneous tumor implantation and adoptive cell transfer were conducted, approximately 40% of mice treated with Scramble T cells developed tumors, vs none in the PHD2/3 KO group. We then subjected tumor-free mice from both groups to an identical tumor cell rechallenge 4 weeks later. The Scramble group demonstrated a tumor recurrence rate of approximately 33%, whereas the PHD2/3 KO group exhibited a mere 5% recurrence rate (Suppl Fig 6B). Furthermore, upon rechallenging mice with intravenous injections of MC38-OVA tumor cells, a notable increase in the expansion of transferred CD8 T cells was observed in the group that received adoptive cell therapy (ACT) of PHD2/3 KO OT-1 T cells compared to the control group (Suppl Fig 6C). This finding provides additional confirmation of the superior “memory” or “progenitor” phenotype exhibited by the PHD2/3 KO T cells (Suppl Fig 6A).”

Supplementary Figure 6: Impact of PHD2/3 deletion in CD8 T cells on their memory phenotype.

(A) Flow cytometry analysis was performed to assess the Slamf6pos and Tim3pos populations as well as the Slamf6neg and Tim3pos populations among CD45.1 positive CD8 T cells within the tumors of mice that received adoptive cell transfer of either Scramble or PHD2/3 KO OT-1 CD8 T cells.

2. Fig 2f – the improvement in cytolytic activity between PHD2/3 KO vs HIF-1 KO is not equivalent regardless of oxygen tension. PHD2/3 KO is universally better at improving CTL activity. This seems to be contrary to the authors suggestion that the effects of PHD2/3 KO are specifically through stabilization of HIF-1a. Can the authors comment on this point?

Author response:

We'd like to provide a clearer explanation of Figure 2f to avoid any misunderstanding. In the normoxic environment, compared to control T cells (depicted by black circles), the absence of PHD2/3 resulted in an enhancement of T cell cytolytic activity (solid blue diamonds), whereas knocking out HIF1 α slightly decreased this function (solid red square).

Under hypoxic conditions, where HIF1 α stabilization occurs, we observed an increased cytolytic activity of T cells (round hollow circle), similar to the increased cytolytic function seen with PHD2/3 knockout in normoxia (solid blue diamonds). However, when HIF1 α was knocked out in the context of hypoxia, the increased cytolytic function of T cells were completely abolished (red hollow square).

Additionally, under hypoxic condition, the PHD2/3 knockout condition led to a further increase of T cell cytolytic activity (blue hollow diamonds), potentially attributable to the further increased HIF1 α stabilization resulting from PHD2/3 deletion under hypoxic conditions (as shown in figure 1b).

These data are consistent with our suggestion that the effects of PHD2/3 KO are through HIF1 α . The formal proof of this is shown in Fig 2j and 2k, with double KO T cells, where the KO of HIF1 α abolished the effects of the PHD2 KO.

3. Some additional controls need to show how PHD2/3 affects specificity: 1) Expression of HIF-2alpha and 2) targets of HIF-1alpha.

Author response:

In accordance with the reviewer's recommendation, we conducted an analysis of the expression levels of HIF-1 alpha target genes. Our results demonstrate a notable increase in the expression of these HIF-1 alpha target genes (*Slc2a1* and *Vegfa*) following PHD2/3 KO (Supplementary Figure 1F and 1G). Furthermore, we investigated the expression of HIF-2 alpha in PHD2/3 KO cells. Contrary to the observed enhancement in HIF-1 alpha expression, our results, as illustrated below and in Supplementary Figure 1L, revealed no significant impact on HIF-2 alpha expression upon PHD2/3 KO. Furthermore, our findings reveal that the increased T cell function achieved through PHD2/3 deletion was completely abolished in HIF-1 α knockout T cells, both in vitro and in vivo. This serves as definitive confirmation of the target specificity of the observed impact of PHD2/3 knockout on T cell function.

These new findings have been incorporated into Supplementary Figure 1, and the corresponding text has been updated in the manuscript between line 126 and line 130, as follows:

“As depicted in Suppl Figure 1F and 1G, the deletion of PHD2/3 leads to an increase in the expression of two HIF-1 α target genes, Slc2a1 and Vegfa. Hypoxia-driven induction of HIF-1 α was lost in HIF-1 α KO cells, while HIF-1 α was clearly stabilized in PHD2/3 KO cells (Figure 1B, Suppl Fig 1H). Notably, the impact of PHD2/3 deletion appears to be specific to HIF-1 α , with no observable alterations observed in HIF-2 α levels (Suppl Figure 1I).”

Supplementary Figure 1F and 1G: RT-qPCR analysis of *Slc2a1* (GLUT1) (F) and *Vegfa* (VEGF) (G) in activated TCRP1A CD8 T cells nucleofected with indicated gRNA-Cas9 complex and incubated under normoxic (21% O₂) or hypoxic (1% O₂) conditions for 24 hours. The mRNA levels of different genes were measured by quantitative RT-qPCR and normalized to *Actb*. Data are expressed as fold change in mRNA copies, with normalization to the condition where scramble T cells were cultured under normoxic conditions.

Supplementary Figure 1I: Representative western blot analysis showing HIF-1α, HIF-2α and Beta-Actin (housekeeping) protein levels in nuclear fractions of scramble and PHD2/3 KO TCRP1A CD8 T cells incubated under normoxic (21% O₂) or hypoxic (1% O₂) conditions for 24 hours.

REVIEWERS' COMMENTS

Reviewer #1 (Remarks to the Author):

the authors have addressed all my concerns and have provided a lot of data to support their conclusions. This is a relevant and very well executed work that should be accepted for publication.

Reviewer #2 (Remarks to the Author):

I have reviewed the revised manuscript provided by the authors. Overall, the changes that have been made address all of my prior concerns. While I acknowledge some challenges with doing metabolomics on the CAR-T cells, this is major point that the authors conclude - PHD2/3 alters metabolism of CAR-T to improve in vivo efficacy. The OVA system is a reasonable "in between" though this model antigen has its own caveats. Somewhere in lines 249-265, the authors should state this limitation.

REVIEWERS' COMMENTS

Reviewer #1 (Remarks to the Author):

the authors have addressed all my concerns and have provided a lot of data to support their conclusions. This is a relevant and very well executed work that should be accepted for publication.

Author response: We would like to thank the reviewer for the positive feedback and valuable comments, which have significantly improved our manuscript.

Reviewer #2 (Remarks to the Author):

I have reviewed the revised manuscript provided by the authors. Overall, the changes that have been made address all of my prior concerns. While I acknowledge some challenges with doing metabolomics on the CAR-T cells, this is major point that the authors conclude - PHD2/3 alters metabolism of CAR-T to improve in vivo efficacy. The OVA system is a reasonable "in between" though this model antigen has its own caveats. Somewhere in lines 249-265, the authors should state this limitation.

Author response: We would like to thank the reviewer for all the valuable comments, which have significantly improved our manuscript.

We recognize that OVA is a model antigen, which is why we primarily used the P1A system, which is a naturally expressed tumor antigen. Our study focused on TCR-based T-cell therapy, prompting our choice of these two models. We did not study CAR T cells. We believe our findings could also apply to CAR-T therapy, but this still requires verification, as we make it clear in lines 406-408, as follows:

“This targeted approach holds promise for optimizing the efficacy and safety of T cell-based immunotherapies and warrants further investigation in preclinical and clinical settings, not only for TCR-based therapy as discussed here, but also potentially for CAR-T cell therapies.

Regarding the OVA system, we have clarified this point, by adding the following sentence in the discussion (lines 342-343):

“We have used CD8 T cells targeting OVA as a model antigen, but also P1A as a more relevant natural tumor antigen.”